# QUADRATIC MODELS FOR UNDERSTANDING NEURAL NETWORK DYNAMICS

## ABSTRACT

In this work, we show that recently proposed quadratic models capture optimization and generalization properties of wide neural networks that cannot be captured by linear models. In particular, we prove that quadratic models for shallow ReLU networks exhibit the "catapult phase" from Lewkowycz et al. (2020) that arises when training such models with large learning rates. We then empirically show that the behaviour of quadratic models parallels that of neural networks in generalization, especially in the catapult phase regime. Our analysis further demonstrates that quadratic models are an effective tool for analysis of neural networks.

## 1 INTRODUCTION

A recent remarkable finding on neural networks, originating from Jacot et al. (2018) and termed as the "transition to linearity" (Liu et al., 2020), is that, as network width goes to infinity, such models become linear functions in the parameter space. Thus, a linear (in parameters) model can be built to accurately approximate wide neural networks under certain conditions. While this finding has helped improve our understanding of trained neural networks (Du et al., 2019; Nichani et al., 2021; Zou & Gu, 2019; Montanari & Zhong, 2020; Ji & Telgarsky, 2019; Chizat et al., 2019), not all properties of finite width neural networks can be understood in terms of linear models, as is shown in several recent works (Yang & Hu, 2020; Ortiz-Jiménez et al., 2021; Long, 2021; Fort et al., 2020). In this work, we show that properties of finitely wide neural networks in optimization and generalization that cannot be captured by linear models are, in fact, manifested in quadratic models.

The training dynamics of linear models with respect to the choice of the learning rates[1] are well-understood (Polyak, 1987). Indeed, such models exhibit *linear* training dynamics, i.e., there exists a critical learning rate, $\eta_{\text{crit}}$, such that the loss converges monotonically if and only if the learning rate is smaller than $\eta_{\text{crit}}$ (see Figure 1a).

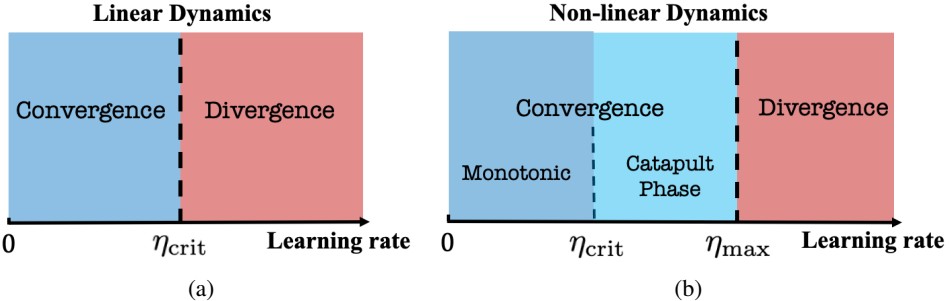

Figure 1: **Optimization dynamics for linear and non-linear models based on choice of learning rate.** (**a**) Linear models either converge monotonically if learning rate is less than $\eta_{\text{crit}}$ and diverge otherwise. (**b**) Unlike linear models, *finitely wide neural networks* and *NQMs Eq. (2) (or general quadratic models Eq. (3))* can additionally observe a catapult phase when $\eta_{\text{crit}} < \eta < \eta_{\text{max}}$.

---

[1]Unless stated otherwise, we always consider the setting where models are trained with squared loss using gradient descent.

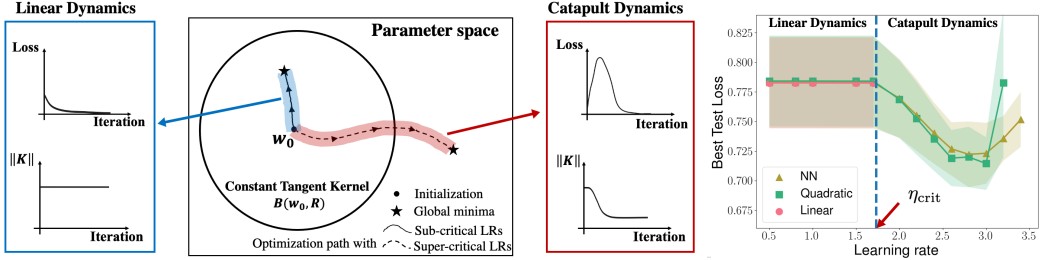

(a) Optimization dynamics for $f$ (wide neural networks): linear dynamics and catapult dynamics.

(b) Generalization performance for $f$, $f_{\text{lin}}$ and $f_{\text{quad}}$.

Figure 2: **(a) Optimization dynamics of wide neural networks with sub-critical and super-critical learning rates.** With sub-critical learning rates ($0 < \eta < \eta_{\text{crit}}$), the tangent kernel of wide neural networks is nearly constant during training, and the loss decreases monotonically. The whole optimization path is contained in the ball $B(\mathbf{w}_0, R) := \{\mathbf{w} : \|\mathbf{w} - \mathbf{w}_0\| \leq R\}$ with a finite radius $R$. With super-critical learning rates ($\eta_{\text{crit}} < \eta < \eta_{\text{max}}$), the catapult phase happens: the loss first increases and then decreases, along with a decrease of the norm of the tangent kernel . The optimization path goes beyond the finite radius ball. **(b) Test loss of $f_{\text{quad}}$, $f$ and $f_{\text{lin}}$ plotted against different learning rates.** With sub-critical learning rates, all three models have nearly identical test loss for any sub-critical learning rate. With super-critical learning rates, $f$ and $f_{\text{quad}}$ have smaller best test loss than the one with sub-critical learning rates. Experimental details are in Appendix J.4.

Recent work Lee et al. (2019) showed that the training dynamics of a wide neural network $f(\mathbf{w}; \boldsymbol{x})$ can be accurately approximated by that of a linear model $f_{\text{lin}}(\mathbf{w}; \boldsymbol{x})$:

$$f_{\text{lin}}(\mathbf{w}; \boldsymbol{x}) = f(\mathbf{w}_0; \boldsymbol{x}) + (\mathbf{w} - \mathbf{w}_0)^T \nabla f(\mathbf{w}_0; \boldsymbol{x}), \tag{1}$$

where $\nabla f(\mathbf{w}_0; \boldsymbol{x})$ denotes the gradient[2] of $f$ with respect to trainable parameters $\mathbf{w}$ at an initial point $\mathbf{w}_0$ and input sample $\boldsymbol{x}$. This approximation holds for learning rates less than $\eta_{\text{crit}} \approx 2/\|\nabla f(\mathbf{w}_0; \boldsymbol{x})\|^2$, when the width is sufficiently large.

However, the training dynamics of finite width neural networks, $f$, can sharply differ from those of linear models when using large learning rates. A striking non-linear property of wide neural networks discovered in Lewkowycz et al. (2020) is that when the learning rate is larger than $\eta_{\text{crit}}$ but smaller than a certain maximum learning rate, $\eta_{\text{max}}$, gradient descent still converges but experiences a "catapult phase." Specifically, the loss initially grows exponentially and then decreases after reaching a large value, along with the decrease of the norm of tangent kernel (see Figure 2a), and therefore, such training dynamics are *non-linear* (see Figure 1b).

As linear models cannot exhibit such a catapult phase, under what models and conditions does this phenomenon arise? The work of Lewkowycz et al. (2020) first observed the catapult phase phenomenon in finite width neural networks and analyzed this phenomenon for a two-layer linear neural network. However, a theoretical understanding of this phenomenon for general non-linear neural networks remains open. In this work, we utilize a quadratic model as a tool to shed light on the optimization and generalization discrepancies between finite and infinite width neural networks. We call this model *Neural Quadratic Model (NQM)* as it is given by the second order Taylor series expansion of $f(\mathbf{w}; \boldsymbol{x})$ around the point $\mathbf{w}_0$:

$$f_{\text{quad}}(\mathbf{w}) = \underbrace{f(\mathbf{w}_0) + (\mathbf{w} - \mathbf{w}_0)^T \nabla f(\mathbf{w}_0)}_{f_{\text{lin}(\mathbf{w})}} + \frac{1}{2}(\mathbf{w} - \mathbf{w}_0)^T H_f(\mathbf{w}_0)(\mathbf{w} - \mathbf{w}_0). \tag{2}$$

Here in the notation we suppress the dependence on the input data $\boldsymbol{x}$, and $H_f(\mathbf{w}_0)$ is the Hessian of $f$ with respect to $\mathbf{w}$ evaluated at $\mathbf{w}_0$.

Indeed, we note that NQMs are contained in a more general class of quadratic models:

$$g(\mathbf{w}; \boldsymbol{x}) = \mathbf{w}^T \phi(\boldsymbol{x}) + \frac{1}{2}\gamma \mathbf{w}^T \Sigma(\boldsymbol{x})\mathbf{w}, \tag{3}$$

---

[2]For non-differentiable functions, e.g. neural networks with ReLU activation functions, we define the gradient based on the update rule used in practice. Similarly, we use $H_f$ to denote the second derivative of $f$ in Eq. (2).

where $\mathbf{w}$ are trainable parameters and $\boldsymbol{x}$ is input data. We discuss the optimization dynamics of such general quadratic models in Section 3.3 and show empirically that they exhibit the catapult phase phenomenon in Appendix J.3. Note that the two-layer linear network analyzed in Lewkowycz et al. (2020) is a special case of Eq. (3), when $\phi(\boldsymbol{x}) = 0$ (See Appendix I).

**Main Contributions.** We prove that NQMs, $f_{\mathrm{quad}}$, which approximate shallow fully-connected ReLU activated neural networks, exhibit catapult phase dynamics. Specifically, we analyze the optimization dynamics of $f_{\mathrm{quad}}$ by deriving the evolution of $f_{\mathrm{quad}}$ and the tangent kernel during gradient descent with squared loss, for a single training example and multiple uni-dimensional training examples. We identify three learning rate regimes yielding different optimization dynamics for $f_{\mathrm{quad}}$, which are (1) converging monotonically (linear dynamics); (2) converging via a catapult phase (catapult dynamics); and (3) diverging. We provide a number of experimental results corroborating our theoretical analysis (See Section 3).

We then empirically show that NQMs, for the architectures of shallow (see Figure 2b as an example) and deep networks, have better test performances when catapult dynamics happens. While this was observed for some synthetic examples of neural networks in Lewkowycz et al. (2020), we systematically demonstrate the improved generalization of NQMs across a range of experimental settings. Namely, we consider fully-connected and convolutional neural networks with ReLU and other activation functions trained with GD/SGD on multiple vision, speech and text datatsets (See Section 4).

To the best of our knowledge, our work is the first to analyze the non-linear wide neural networks in the catapult regime through the perspective of the quadratic approximation. While NQMs (or quadratic models) were proposed and analyzed in Roberts et al. (2022), our work focuses on the properties of NQMs in the large learning rate regime, which has not been discussed in Roberts et al. (2022). Similarly, the following related works did not study catapult dynamics. Huang & Yau (2020) analyzed higher order approximations to neural networks under gradient flow (infinitesimal learning rates). Bai & Lee (2019) studied different quadratic models with randomized second order terms and Zhang et al. (2019) considered the loss in the quadratic form, where no catapult phase happens.

**Discontinuity in dynamics transition.** In the ball $B(\mathbf{w}_0, R) := \{\mathbf{w} : \|\mathbf{w} - \mathbf{w}_0\| \leq R\}$ with constant radius $R > 0$, the transition to linearity of a wide neural network (with linear output layer) is continuous in the network width $m$. That is, the deviation from the network function to its linear approximation within the ball can be continuously controlled by the Hessian of the network function, i.e. $H_f$, which scales with $m$ (Liu et al., 2020):

$$\|f(\mathbf{w}) - f_{\mathrm{lin}}(\mathbf{w})\| \leq \sup_{\mathbf{w} \in B(\mathbf{w}_0, R)} \|H_f(\mathbf{w})\| R^2 = \tilde{O}(1/\sqrt{m}). \tag{4}$$

Using the inequality from Eq. (4), we obtain $\|f_{\mathrm{quad}} - f_{\mathrm{lin}}\| = \tilde{O}(1/\sqrt{m})$, hence $f_{\mathrm{quad}}$ transitions to linearity continuously as well in $B(\mathbf{w}_0, R)^3$. Given the continuous nature of the transition to linearity, one may expect that the transition from non-linear dynamics to linear dynamics for $f$ and $f_{\mathrm{quad}}$ is continuous in $m$ as well. Namely, one would expect that the domain of catapult dynamics, $[\eta_{\mathrm{crit}}, \eta_{\mathrm{max}}]$, shrinks and ultimately converges to a single point, i.e., $\eta_{\mathrm{crit}} = \eta_{\mathrm{max}}$, as $m$ goes to infinity, with non-linear dynamics turning into linear dynamics. However, as shown both analytically and empirically, the transition is *not* continuous, for both network functions $f$ and NQMs $f_{\mathrm{quad}}$, since the domain of the catapult dynamics can be independent of the width $m$ (or $\gamma$). Additionally, the length of the optimization path of $f$ in catapult dynamics grows with $m$ since otherwise, the optimization path could be contained in a ball with a constant radius independent of $m$, in which $f$ can be approximated by $f_{\mathrm{lin}}$. Since $f_{\mathrm{lin}}$ diverges in catapult dynamics, by the approximation, $f$ diverges as well, which contradicts the fact that $f$ can converge in catapult dynamics (See Figure 2a).

## 2 NOTATION AND PRELIMINARY

We use bold lowercase letters to denote vectors and capital letters to denote matrices. We denote the set $\{1, 2, \cdots, n\}$ by $[n]$. We use $\|\cdot\|$ to denote the Euclidean norm for vectors and the spectral norm for matrices. We use $\odot$ to denote element-wise multiplication (Hadamard product) for vectors. We use $\lambda_{\max}(A)$ and $\lambda_{\min}(A)$ to denote the largest and smallest eigenvalue of a matrix $A$, respectively.

---

[3]For general quadratic models in Eq. (3), the transition to linearity is continuously controlled by $\gamma$.

Given a model $f(\mathbf{w}; \boldsymbol{x})$, where $\boldsymbol{x}$ is input data and $\mathbf{w}$ are model parameters, we use $\nabla_{\mathbf{w}} f$ to represent the partial first derivative $\partial f(\mathbf{w}; \boldsymbol{x})/\partial \mathbf{w}$. When clear from context, we let $\nabla f := \nabla_{\mathbf{w}} f$ for ease of notation. We use $H_f$ and $H_{\mathcal{L}}$ to denote the Hessian (second derivative matrix) of the function $f(\mathbf{w}; \boldsymbol{x})$ and the loss $\mathcal{L}(\mathbf{w})$ with respect to parameters $\mathbf{w}$, respectively.

In the paper, we consider the following supervised learning task: given training data $\{(\boldsymbol{x}_i, y_i)\}_{i=1}^n$ with data $\boldsymbol{x}_i \in \mathbb{R}^d$ and labels $y_i \in \mathbb{R}$ for $i \in [n]$, we minimize the empirical risk with the squared loss $\mathcal{L}(\mathbf{w}) = \frac{1}{2} \sum_{i=1}^n (f(\mathbf{w}; \boldsymbol{x}_i) - y_i)^2$. Here $f(\mathbf{w}; \cdot)$ is a parametric family of models, e.g., a neural network or a kernel machine, with parameters $\mathbf{w} \in \mathbb{R}^p$. We use full-batch gradient descent to minimize the loss, and we denote trainable parameters $\mathbf{w}$ at iteration $t$ by $\mathbf{w}(t)$. With constant step size (learning rate) $\eta$, the update rule for the parameters is:

$$\mathbf{w}(t + 1) = \mathbf{w}(t) - \eta \frac{d\mathcal{L}(\mathbf{w})}{d\mathbf{w}}(t), \quad \forall t \geq 0.$$

**Definition 1** (Tangent Kernel). The tangent kernel $K(\mathbf{w}; \cdot, \cdot)$ of $f(\mathbf{w}; \cdot)$ is defined as

$$K(\mathbf{w}; \boldsymbol{x}, \boldsymbol{z}) = \langle \nabla f(\mathbf{w}; \boldsymbol{x}), \nabla f(\mathbf{w}; \boldsymbol{z}) \rangle, \quad \forall \boldsymbol{x}, \boldsymbol{z} \in \mathbb{R}^d. \tag{5}$$

In the context of the optimization problem with $n$ training examples, the tangent kernel matrix $K \in \mathbb{R}^{n \times n}$ satisfies $K_{i,j}(\mathbf{w}) = K(\mathbf{w}; \boldsymbol{x}_i, \boldsymbol{x}_j)$, $i, j \in [n]$. The critical learning rate for optimization is given as follows.

**Definition 2** (Critical learning rate). With an initialization of parameters $\mathbf{w}_0$, the critical learning rate of $f(\mathbf{w}; \cdot)$ is defined as

$$\eta_{\text{crit}} := 2/\lambda_{\max}(H_{\mathcal{L}}(\mathbf{w}_0)). \tag{6}$$

A learning rate $\eta$ is said to be *sub-critical* if $0 < \eta < \eta_{\text{crit}}$ or *super-critical* if $\eta_{\text{crit}} < \eta < \eta_{\max}$. Here $\eta_{\max}$ is the maximum leaning rate such that the optimization of $\mathcal{L}(\mathbf{w})$ initialized at $\mathbf{w}_0$ can converge.

**Dynamics for Linear models.** When $f$ is linear in $\mathbf{w}$, the gradient, $\nabla f$, and tangent kernel are constant: $K(\mathbf{w}(t)) = K(\mathbf{w}_0)$. Therefore, gradient descent dynamics are:

$$F(\mathbf{w}(t + 1)) - \mathbf{y} = (I - \eta K(\mathbf{w}_0))(F(\mathbf{w}(t)) - \mathbf{y}), \quad \forall t \geq 0, \tag{7}$$

where $F(\mathbf{w}_0) = [f_1(\mathbf{w}_0), ..., f_n(\mathbf{w}_0)]^T$ with $f_i(\mathbf{w}_0) = f(\mathbf{w}_0; \boldsymbol{x}_i)$.

Noting that $H_{\mathcal{L}}(\mathbf{w}_0) = \nabla F(\mathbf{w}_0)^T \nabla F(\mathbf{w}_0)$ and that tangent kernel $K(\mathbf{w}_0) = \nabla F(\mathbf{w}_0) \nabla F(\mathbf{w}_0)^T$ share the same positive eigenvalues, we have $\lambda_{\max}(H_{\mathcal{L}}(\mathbf{w}_0)) = \lambda_{\max}(K(\mathbf{w}_0))$, and hence,

$$\eta_{\text{crit}} = 2/\lambda_{\max}(K(\mathbf{w}_0)). \tag{8}$$

Therefore, from Eq. equation 7, if $0 < \eta < \eta_{\text{crit}}$, the loss $\mathcal{L}$ decreases monotonically and if $\eta > \eta_{\text{crit}}$, the loss $\mathcal{L}$ diverges. Note that the critical and maximum learning rates are equal in this setting.

## 3 OPTIMIZATION DYNAMICS IN NEURAL QUADRATIC MODELS

In this section, we analyze the gradient descent dynamics of the NQM that approximates two-layer fully connected ReLU activated neural networks up to the second order. We show that the extra quadratic term in NQMs allows for catapult convergence: the loss increases at early stage and then converges afterwards. Note that this type of convergence happens with super-critical learning rates and cannot happen for linear models. Interestingly, the top eigenvalues of the tangent kernel typically decrease after the catapult phase, while they are nearly constant when training with sub-critical learning rates, where the loss converges monotonically.

**Neural Quadratic Model (NQM).** Consider the NQM that approximates the following two-layer neural network:

$$f(\mathbf{u}, \mathbf{v}; \boldsymbol{x}) = \frac{1}{\sqrt{m}} \sum_{i=1}^m v_i \sigma\left(\frac{1}{\sqrt{d}} \mathbf{u}_i^T \boldsymbol{x}\right), \tag{9}$$

where $\mathbf{u}_i \in \mathbb{R}^d$, $v_i \in \mathbb{R}$ for $i \in [m]$ are trainable parameters, $\boldsymbol{x} \in \mathbb{R}^d$ is the input, and $\sigma(\cdot)$ is the ReLU activation function. Each parameter is initialized i.i.d. following the standard normal

distribution, $\mathcal{N}(0,1)$. Letting $g(\mathbf{u}, \mathbf{v}; \boldsymbol{x}) := f_{\text{quad}}(\mathbf{u}, \mathbf{v}; \boldsymbol{x})$, this NQM has the following expression (See the full derivation in Appendix A):

$$g(\mathbf{u}, \mathbf{v}; \boldsymbol{x}) = f(\mathbf{u}_0, \mathbf{v}_0; \boldsymbol{x}) + \frac{1}{\sqrt{md}} \sum_{i=1}^{m} v_{0,i}(\mathbf{u}_i - \mathbf{u}_{0,i})^T \boldsymbol{x} \mathbb{1}_{\left\{\mathbf{u}_{0,i}^T \boldsymbol{x} \geq 0\right\}} + \frac{1}{\sqrt{md}} \sum_{i=1}^{m} (v_i - v_{0,i}) \sigma\left(\mathbf{u}_{0,i}^T \boldsymbol{x}\right)$$

$$+ \frac{1}{\sqrt{md}} \sum_{i=1}^{m} (v_i - v_{0,i})(\mathbf{u}_i - \mathbf{u}_{0,i})^T \boldsymbol{x} \mathbb{1}_{\left\{\mathbf{u}_{0,i}^T \boldsymbol{x} \geq 0\right\}}. \tag{10}$$

Given training data $\{\boldsymbol{x}_i, y_i\}_{i=1}^{n}$, we minimize the empirical risk with the squared loss $\mathcal{L}(\mathbf{w}) = \frac{1}{2} \sum_{i=1}^{n} (g(\mathbf{w}; \boldsymbol{x}_i) - y_i)^2$ using GD with constant learning rate $\eta$. Throughout this section, we denote $g(\mathbf{u}(t), \mathbf{v}(t); \boldsymbol{x})$ by $g(t)$ and its tangent kernel $K(\mathbf{u}(t), \mathbf{v}(t))$ by $K(t)$, where $t$ is the iteration of GD. We assume $\|\boldsymbol{x}_i\| = O(1)$ and $|y_i| = O(1)$ for $i \in [n]$, and we assume the width of $f$ is much larger than the input dimension $d$ and the data size $n$, i.e., $m \gg \max\{d, n\}$. Hence, $d$ and $n$ can be regarded as small constants. In the whole paper, we use the big-O and small-o notation with respect to the width $m$. Below, we start with the single training example case, which already showcases the non-linear dynamics of NQMs.

## 3.1 CATAPULT DYNAMICS WITH SINGLE TRAINING EXAMPLE

In this subsection, we consider training dynamics of NQMs with a single training example $(\boldsymbol{x}, y)$ where $\boldsymbol{x} \in \mathbb{R}^d$ and $y \in \mathbb{R}$. In this case, the tangent kernel matrix $K$ reduces to a scalar, and we denote $K$ by $\lambda$ to distinguish it from a matrix.

By gradient descent with step size $\eta$, the updates for $g(t) - y$ and $\lambda(t)$, which we refer to as dynamics equations, can be derived as follows (see Appendix B.1):

**Dynamics equations.**

$$g(t+1) - y = \left(1 - \eta\lambda(t) + \underbrace{\frac{\|\boldsymbol{x}\|^2}{md}\eta^2(g(t) - y)g(t)}_{R_g(t)}\right)(g(t) - y) := \mu(t)(g(t) - y), \tag{11}$$

$$\lambda(t+1) = \lambda(t) + \underbrace{\eta\frac{\|\boldsymbol{x}\|^2}{md}(g(t) - y)^2\left(\eta\lambda(t) - 4\frac{g(t)}{g(t) - y}\right)}_{R_\lambda(t)}, \quad \forall t \geq 0. \tag{12}$$

Note that as the loss is given by $\mathcal{L}(t) = 1/2(g(t) - y)^2$, to understand convergence, it suffices to analyze the dynamics equations above. Compared to the linear dynamics Eq. (7), this non-linear dynamics has extra terms $R_g(t)$ and $R_\lambda(t)$, which are induced by the non-linear term in the NQM. We will see that the convergence of gradient descent depends on the scale and sign of $R_g(t)$ and $R_\lambda(t)$. For example, for constant learning rate that is slightly larger than $\eta_{\text{crit}}$ (which would result in divergence for linear models), $R_\lambda(t)$ stays negative during training, resulting in both monotonic decrease of tangent kernel $\lambda$ and convergence of the loss.

For the scale of $\lambda_0$, which is non-negative by Definition 1, we can show that with high probability over random initialization, $|\lambda_0| = \Theta(1)$ (see Appendix D). As $\lambda(t) = \lambda_0 + \sum_{\tau=0}^{t} R_\lambda(\tau)$, to track the scale of $|\mu(t)|$, we will focus on the scale and sign of $R_g(t)$ and $R_\lambda(t)$ in the following analysis. We start by establishing monotonic convergence for sub-critical learning rates.

**Monotonic convergence: sub-critical learning rates** ($\eta < 2/\lambda(0)$). The key observation we use is that when $|g(t)|$ is small, i.e., of the order $o(\sqrt{m})$, and $\lambda(t) = \Theta(1)$, $|R_g(t)|$ and $|R_\lambda(t)|$ are of the order $o(1)$(see Proposition 3 in Appendix F). Then, the dynamics equations approximately reduce to the ones of linear dynamics:

$$g(t+1) - y = (1 - \eta\lambda(t) + o(1))(g(t) - y),$$
$$\lambda(t+1) = \lambda(t) + o(1).$$

Note that at initialization, with high probability over random initialization, the output satisfies $|g(0)| = O(1)$ if $\|\boldsymbol{x}\| = O(1)$ (Jacot et al., 2018), and we have shown $\lambda(0) = \Theta(1)$. Applying

Proposition 3, with the choice of $\eta$, we have for all $t \geq 0$, $|\mu(t)| = |1 - \eta\lambda(t) + o(1)| < 1$; hence, $|g(t) - y|$ decreases monotonically. The cumulative change on the tangent kernel will be $o(1)$, i.e., $\sum_t |R_\lambda(t)| = o(1)$, since for all $t$, $|R_\lambda(t)| = o(1)$ and the loss decreases exponentially hence $\sum \|R_K(t)\| = o(1) \cdot \log O(1) = o(1)$.

**Catapult convergence: super-critical learning rates** $(2/\lambda(0) < \eta < 4/\lambda(0))$. The training dynamics are given by the following theorem.

**Theorem 1** (Catapult dynamics on a single training example). *Consider training a NQM, Eq. (10), with squared loss on a single training example by gradient descent. If the learning rate is super-critical, i.e., $2/\lambda_0 < \eta < 4/\lambda_0$, then there exist $T_1, T_2, T_3, T_4$ such that $0 < T_1 < T_2 < T_3 < T_4$ and the training dynamics exhibits:*

*(i)* **Increasing phase:** $t \in [0, T_1]$. *In this phase, $\mathcal{L}(t) = o(m)$. The loss grows exponentially and the tangent kernel is nearly constant, i.e. $|\lambda(t) - \lambda(0)| = o(1)$.*

*(ii)* **Peak phase:** $t \in [T_2, T_3]$. *In this phase, $\mathcal{L}(t) = \Theta(m)$. The tangent kernel decreases significantly: $\lambda(t+1) - \lambda(t) < 0$ and $|\lambda(t+1) - \lambda(t)| = \Theta(1)$.*

*(iii)* **Decreasing phase:** $t \in [T_4, \infty)$. *In this phase, the loss satisfies $\mathcal{L}(t) = o(m)$ again and decreases. The tangent kernel is nearly constant until convergence: $|\lambda(t) - \lambda(\infty)| = o(1)$.*

*Furthermore, $T_1 = o(\log m)$, $T_2 = \Theta(\log m)$, $T_3 - T_2 = \Theta(1)$ and $T_4 = \Theta(\log m)$.*

*Proof of Theorem 1.* We will analyze the training dynamics in each phase sequentially.

*(i) Increasing phase.* At the beginning, $|g(t)|$ grows exponentially following linear dynamics. Specifically, since $|g(0)| = O(1)$, by Proposition 3, we have $|R_g(0)| = o(1)$ and $|R_\lambda(0)| = o(1)$. The loss grows exponentially at iteration 0: by the choice of the learning rate, $\mu(0)$ satisfies

$$|\mu(0)| = |1 - \eta\lambda(0) + R_g(0)| = |1 - \eta\lambda(0) + o(1)| > 1.$$

Therefore $|g(1) - y| > |g(0) - y|$, and the tangent kernel almost does not change: $\lambda(1) = \lambda(0) + o(1)$.

We can recursively apply this argument for the following steps as long as $|g(t)| = o(\sqrt{m})$ according to Proposition 3. Note that in the increasing phase, the loss grows exponentially due to $|\mu(t)| > 1$. Therefore, the tangent kernel is nearly constant since the cumulative change is $\sum_{t=0}^{T_1} |R_\lambda(t)| = \sum_{t=0}^{T_1} \Theta\left(g(t)^2/m\right) = o(1)$, where we use the fact that $g(t)^2$ grows exponentially to the order of $o(m)$. And we can get $T_1 = o(\log m)$.

Furthermore, it is not hard to see that until the loss grows to the order of $\Theta(m)$, the tangent kernel does not change much hence the loss keeps increasing exponentially.

*(ii) Peak phase.* The key observation we use here is that when $|g(t)|$ is large, i.e., of the order $\Theta(\sqrt{m})$, $|R_g(t)|$ and $|R_\lambda(t)|$ will be of the order $\Theta(1)$(see Proposition 4 in Appendix F), which can lead to the decrease of the loss.

In the peak phase, we have $|g(t)| = \Theta(\sqrt{m})$, then by Proposition 4, the scale of $|R_\lambda(t)|$ is $\Theta(1)$, and $R_\lambda(t) < 0$ since $\lambda(t) < 4/\eta$ (in the increasing phase, $\lambda(t)$ almost does not change, hence we have $\lambda(t) \approx \lambda(0) < 4/\eta$ before the peak phase, and will decrease significantly). Consequently, by Eq. (12), $\lambda(t)$ will have significant decrease as $|R_\lambda(t)| = \Theta(1)$ and $\lambda(t+1) = \lambda(t) + R_\lambda(t) < \lambda(t)$, which is further smaller than $4/\eta$.

Similarly, when $|g(t)| = \Theta(\sqrt{m})$, $|R_g(t)| = \Theta(1)$ and $R_g(t) > 0$ by Proposition 4. Then we can see the increase of loss slows down compared to that in the increasing phase:

$$|\mu(t)| = |1 - \eta\lambda(t) + R_g(t)| < |1 - \eta\lambda(0) + R_g(0)| = |1 - \eta\lambda(0) + o(1)| \approx \mu(0).$$

In general, the loss grows exponentially prior to the peak phase hence $T_2 = \Theta(\log m)$. And the peak phase only lasts $\Theta(1)$ steps during training, i.e., $|T_3 - T_2| = \Theta(1)$, since the decrease of $\lambda(t)$ is $\Theta(1)$ at each step and the training dynamics will enter into the decreasing phase once $|\mu(t)| < 1$, which will happen if $\lambda(t)$ is sufficiently small.

*(iii) Decreasing phase.* The peak phase ends when $|\mu(t)| < 1$, then $|g(t) - y|$ starts to decrease. We note that our analysis implicitly assumes that the optimization path will not arrive at the saddle point, i.e., $|\mu(t)| = 1$, and it is generally true with discrete steps.

Recall that at the peak phase, $\lambda(t)$ decreases, and when $|\mu(t)| > 1$, $|g(t)|$ increases which causes the increase of $R_g(t)$ since $R_g(t)$ scales with $|g(t)|$. As a result $|\mu(t)| = |1 - \eta\lambda(t) + R_g(t)|$ decreases

and will be less than 1 ultimately. Note that though $|g(t) - y|$ starts to decrease, it is still of the order $\Theta(\sqrt{m})$, which makes $\lambda(t)$ decrease significantly as $R_\lambda(t) = \Theta(1)$ by Proposition 4.

The decrease of $\lambda(t)$ will stop once $|g(t)|$ decreases to the order of $o(m)$, as $R_\lambda(t) = o(1)$ again by Proposition 3. At that moment, $T_4 - T_3 = \Theta(\log m)$ as the loss decreases exponentially and the linear dynamics dominate again. Therefore, similar to the increasing phase, starting from $t$ such that $|g(t)| = o(\sqrt{m})$, the loss decreases exponentially hence the change of $\lambda(t)$ until convergence is $o(1)$.

$\square$

**Divergence ($\eta > \eta_{\max} = 4/\lambda(0)$).** Initially, it follows the same dynamics with those in the increasing phase in catapult convergence: the loss increases exponentially and the tangent kernel is nearly constant. However, when $|g(t)|$ grows to the order of $\Theta(\sqrt{m})$, corresponding to the peak phase in catapult convergence, $\lambda(t)$ does not decrease but increases significantly. Specifically, since $\eta > 4/\lambda(0)$, we approximately have $\eta > 4/\lambda(t)$ at the end of the increasing phase by the same analysis in catapult convergence. By Proposition 3, $R_\lambda(t) > 0$, then $\lambda(t)$ increases as $\lambda(t+1) = \lambda(t) + R_\lambda(t) > \lambda(t)$. Larger $\lambda(t)$ leads to the faster increase on $\lambda(t)$, hence $|\mu(t)|$ becomes even larger. As a result, $|g(t) - y|$ grows faster, therefore the loss diverges.

## 3.2 CATAPULT DYNAMICS WITH MULTIPLE TRAINING EXAMPLES

In this subsection we show the catapult phase will happen for NQMs Eq. (9) with multiple training examples. We assume unidimensional input data, which is common in the literature and simplifies the analysis for neural networks (see for example Williams et al. (2019); Savarese et al. (2019)).

**Assumption 1.** The input dimension $d = 1$ and not all $x_i$ is 0, i.e., $\sum |x_i| > 0$.

Since $x_i$ is a scalar for all $i \in [n]$, with the homogeneity of ReLU activation function, we can compute the exact eigenvectors of $K(t)$ for all $t \geq 0$. To that end, we group the data into two sets $\mathcal{S}_+$ and $\mathcal{S}_-$ according to their sign:

$$\mathcal{S}_+ := \{i : x_i \geq 0, i \in [n]\}, \quad \mathcal{S}_- := \{i : x_i < 0, i \in [n]\}.$$

Now we have the proposition for the tangent kernel $K$ (the proof is deferred to Appendix C):

**Proposition 1** (Eigenvectors and low rank structure of $K$). *For any* $\mathbf{u}, \mathbf{v} \in \mathbb{R}^m$, $\mathrm{rank}(K) \leq 2$. *Furthermore,* $\boldsymbol{p}_1, \boldsymbol{p}_2$ *are eigenvectors of* $K$, *where* $p_{1,i} = x_i \mathbb{1}_{\{i \in \mathcal{S}_+\}}$, $p_{2,i} = x_i \mathbb{1}_{\{i \in \mathcal{S}_-\}}$, *for* $i \in [n]$.

Note that when all $x_i$ are of the same sign, $\mathrm{rank}(K) = 1$ and $K$ only has one eigenvector (either $\boldsymbol{p}_1$ or $\boldsymbol{p}_2$ depending on the sign). It is in fact a simpler setting since we only need to consider one direction, whose analysis is covered by the one for $\mathrm{rank}(K) = 2$. Therefore, in the following we will assume $\mathrm{rank}(K) = 2$. We denote two eigenvalues of $K(t)$ by $\lambda_1(t)$ and $\lambda_2(t)$ corresponding to $\boldsymbol{p}_1$ and $\boldsymbol{p}_2$ respectively, i.e., $K(t)\boldsymbol{p}_1 = \lambda_1(t)\boldsymbol{p}_1$, $K(t)\boldsymbol{p}_2 = \lambda_2(t)\boldsymbol{p}_2$. Without loss of generality, we assume $\lambda_1(0) \geq \lambda_2(0)$.

We similarly analyze the dynamics equations for multiple training examples (see Eq. (14) and (15) which are update equations of $\mathbf{g}(t) - \mathbf{y}$ and $K(t)$) with different learning rates. And we formulate the result for the catapult dynamics, which happens when training with super-critical learning rates, into the following theorem:

**Theorem 2** (Catapult dynamics on multiple training examples). *Consider training a NQM Eq. (10) with squared loss on multiple training examples by gradient descent. Under Assumption 1, if the learning rate is super-critical i.e.,* $2/\lambda_1(0) < \eta < \min\{2/\lambda_2(0), 4/\lambda_1(0)\}$, *then there exist* $T_1, T_2, T_3, T_4$ *such that* $0 < T_1 < T_2 < T_3 < T_4$ *and the training dynamics exhibits:*

(i) **Increasing phase:** $t \in [0, T_1]$. *In this phase,* $\mathcal{L}(t) = o(m)$. *The loss grows exponentially; both eigenvalues are nearly constant i.e.,* $|\lambda_k(t) - \lambda_k(0)| = o(1)$ *for* $k = 1, 2$.

(ii) **Peak phase:** $t \in [T_2, T_3]$. *In this phase,* $\mathcal{L}(t) = \Theta(m)$. *For the tangent kernel:*

(a) *If* $2/\lambda_2(0) \leq 4/\lambda_1(0)$, *both eigenvalues decrease significantly, i.e.,* $\lambda_1(t+1) - \lambda_1(t) < 0$, $\lambda_2(t+1) - \lambda_2(t) < 0$ *and both difference are of the order* $\Theta(1)$.

(b) *If* $2/\lambda_2(0) > 4/\lambda_1(0)$, *only* $\lambda_1(t)$ *decreases significantly.*

(iii) **Decreasing phase:** $t \in [T_4, \infty)$. *In this phase, the loss satisfies* $\mathcal{L}(t) = o(m)$ *again and decreases. Both eigenvalues are nearly constant until convergence:* $|\lambda_k(t) - \lambda_k(\infty)| = o(1)$, *for* $k = 1, 2$.

*Furthermore, $T_1 = o(\log m)$, $T_2 = \Theta(\log m)$, $T_3 - T_2 = \Theta(1)$ and $T_4 = \Theta(\log m)$.*

The proof idea is similar to the case for a single training example: due to the low rank structure of the tangent kernel, we only need to analyze directions $\boldsymbol{p}_1$ and $\boldsymbol{p}_2$. It turns out that we can analyze the dynamics separately for each direction as we show the training dynamics in these two directions are almost independent to each other. We defer the proof of Theorem 2 and the analysis of other two parts of non-linear dynamics: monotonic convergence and divergence to Appendix H.

We verify the theoretical results above for multiple training examples via the experiments in Figure 3.

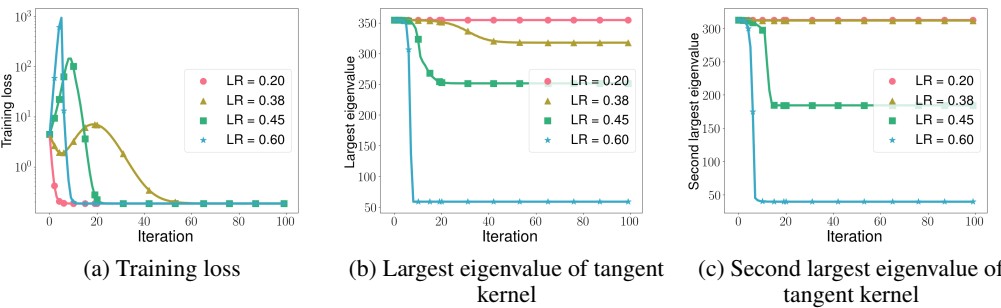

(a) Training loss      (b) Largest eigenvalue of tangent kernel      (c) Second largest eigenvalue of tangent kernel

Figure 3: **Training dynamics of NQMs for multiple examples case with different learning rates.** By our analysis, two critical values are $2/\lambda_1(0) = 0.37$ and $2/\lambda_2(0) = 0.39$. When $\eta < 0.37$, linear dynamics dominate hence the kernel is nearly constant; when $0.37 < \eta < 0.39$, the catapult phase happens in $\boldsymbol{p}_1$ and only $\lambda_1(t)$ decreases; when $0.39 < \eta < \eta_{\max}$, the catapult phase happens in $\boldsymbol{p}_1$ and $\boldsymbol{p}_2$ hence both $\lambda_1(t)$ and $\lambda_2(t)$ decreases. The experiment details can be found in Appendix J.1.

## 3.3 CONNECTION TO WIDE NEURAL NETWORKS AND GENERAL QUADRATIC MODELS

**Wide neural networks.** We have seen that NQMs, with fixed Hessian, exhibit the catapult phase phenomenon. Therefore, the change in the Hessian of wide neural networks during training is not required to produce the catapult phase. In our analysis, we show that the catapult phase arises because the eigenvectors of the tangent kernel "align" with the Hessian's spectrum, i.e., $H_{g_i}$ for $i \in S_+$ are proportional with coefficients $\boldsymbol{p}_1$, and the same holds for $H_{g_i}$ for $i \in S_-$ with coefficients $\boldsymbol{p}_2$. E.g., $H_{g_j}/p_{1,j} = H_{g_k}/p_{1,k}$ if $j, k \in S_+$. We believe this idea can be used to analyze the catapult dynamics in wide neural networks with changing Hessian. A similar behaviour of top eigenvalues of the tangent kernel with the one for NQMs is observed for wide neural networks when training with different learning rates (See Figure 5 in Appendix J).

**General quadratic models.** As mentioned in the introduction, NQMs are contained in a general class of quadratic models of the form given in Eq. (3). We show that the two-layer linear neural network analyzed in Lewkowycz et al. (2020) is a special case of Eq. (3), and we provide a more general condition for such models to have catapult dynamics in Appendix I.

Furthermore, we empirically observe that a broader class of quadratic models $g$ can have catapult dynamics simply by letting $\phi(\boldsymbol{x})$ and $\Sigma$ be random and assigning a small value to $\gamma$ (See Appendix J.3).

## 4 QUADRATIC MODELS PARALLEL NEURAL NETWORKS IN GENERALIZATION

In this section, we empirically compare the test performance of three different models considered in this paper upon varying learning rate. In particular, we consider (1) the NQM, $f_{\text{quad}}$; (2) corresponding neural networks, $f$; and (3) the linear model, $f_{\text{lin}}$.

We implement our experiments on 3 vision datasets: CIFAR-2 (a 2-class subset of CIFAR-10 (Krizhevsky et al., 2009)), MNIST (LeCun et al., 1998), and SVHN (The Street View House Numbers) (Netzer et al., 2011), 1 speech dataset: Free Spoken Digit dataset (FSDD) (Jakobovski, 2020) and 1 text dataset: AG NEWS (Gulli, 2005).

In all experiments, we train the models by minimizing the squared loss using standard GD/SGD with constant learning rate $\eta$. We report the best test loss achieved during the training process with

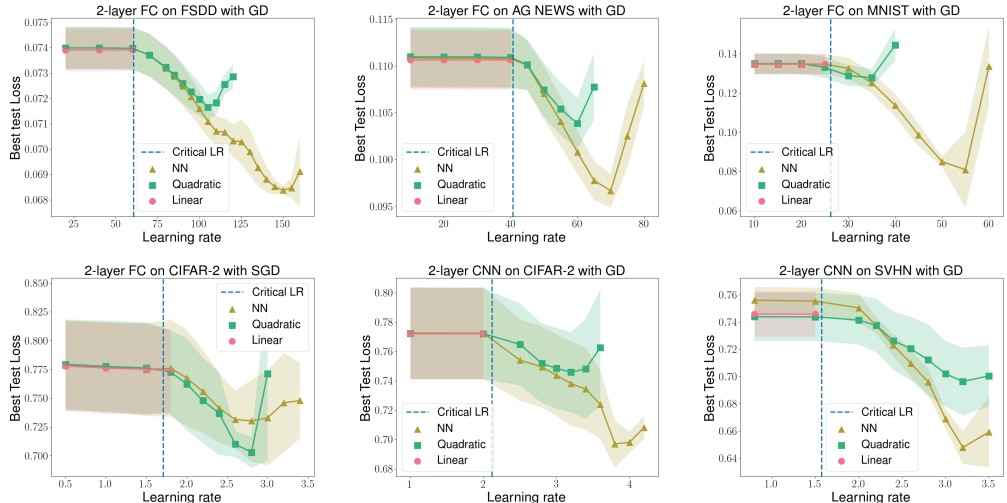

Figure 4: **Best test loss plotted against different learning rates for $f(\mathbf{w})$, $f_{\mathrm{lin}}(\mathbf{w})$ and $f_{\mathrm{quad}}(\mathbf{w})$ across a variety of datasets and network architectures.**

each learning rate. Experimental details can be found in Appendix J.4. We also report the best test accuracy in Appendix J.5. For networks with 3 layers, see Appendix J.6. From the experimental results, we observe the following:

**Sub-critical learning rates.**    In accordance with our theoretical analyses, we observe that all three models have nearly identical test loss for any sub-critical learning rate. Specifically, note that as the width $m$ increases, $f$ and $f_{\mathrm{quad}}$ will transition to linearity in the ball $B(\mathbf{w}_0, R)$:

$$\|f - f_{\mathrm{lin}}\| = \tilde{O}(1/\sqrt{m}), \quad \|f_{\mathrm{quad}} - f_{\mathrm{lin}}\| = \tilde{O}(1/\sqrt{m}),$$

where $R > 0$ is a constant which is large enough to contain the optimization path with respect to sub-critical learning rates. Thus, the generalization performance of these three models will be similar when $m$ is large, as shown in Figure 4.

**Super-critical learning rates.**    The best test loss of both $f(\mathbf{w})$ and $f_{\mathrm{quad}}(\mathbf{w})$ is consistently smaller than the one with sub-critical learning rates, and decreases for an increasing learning rate in a range of values beyond $\eta_{\mathrm{crit}}$, which was observed for wide neural networks in Lewkowycz et al. (2020).

As discussed in the introduction, with super-critical learning rates, both $f_{\mathrm{quad}}$ and $f$ can be observed to have catapult phase, while the loss of $f_{\mathrm{lin}}$ diverges. Together with the similar behaviour of $f_{\mathrm{quad}}$ and $f$ in generalization with super-critical learning rates, we believe NQMs are a better model to understand $f$ in training and testing dynamics, than the linear approximation $f_{\mathrm{lin}}$.

In Figure 4 we report the results for networks with ReLU activation function. We also implement the experiments using networks with Tanh and Swish (Ramachandran et al., 2017) activation functions, and observe the same phenomena in generalization for $f$, $f_{\mathrm{lin}}$ and $f_{\mathrm{quad}}$ (See Appendix J.7).

## 5    CONCLUSIONS

In this paper, we use quadratic models as a tool to better understand optimization and generalization properties of finite width neural networks trained using large learning rates. Notably, we prove that quadratic models exhibit properties of neural networks such as the catapult dynamics that cannot be explained using linear models, which importantly includes linear approximations to neural networks given by the neural tangent kernel. Interestingly, we show empirically that quadratic models mimic the generalization properties of neural networks when trained with large learning rate, and that such models perform better than linearized neural networks.

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

## A  DERIVATION OF NQM

We will derive the NQM that approximate the two-layer fully connected ReLU activated neural networks based on Eq. (2).

The first derivative of $f$ can be computed by:

$$\frac{\partial f}{\partial \mathbf{u}_i} = \frac{1}{\sqrt{md}} v_i \mathbb{1}_{\{\mathbf{u}_i^T \boldsymbol{x} \geq 0\}} \boldsymbol{x}^T, \quad \frac{\partial f}{\partial v_i} = \frac{1}{\sqrt{m}} \sigma\left(\frac{1}{\sqrt{d}}\mathbf{u}_i^T\boldsymbol{x}\right), \quad \forall i \in [m].$$

And each entry of the Hessian of $f$, i.e., $H_f$, can be computed by

$$\frac{\partial^2 f}{\partial \mathbf{u}_i^2} = \mathbf{0}, \ \frac{\partial^2 f}{\partial v_i^2} = 0, \ \frac{\partial^2 f}{\partial \mathbf{u}_i v_i} = \frac{1}{\sqrt{md}} \mathbb{1}_{\{\mathbf{u}_i^T \boldsymbol{x} \geq 0\}} \boldsymbol{x}^T, \quad \forall i \in [m].$$

Now we get $f_{\text{quad}}$ taking the following form

$$\mathbf{NQM}: f_{\text{quad}}(\mathbf{u}, \mathbf{v}; \boldsymbol{x}) = f(\mathbf{u}_0, \mathbf{v}_0; \boldsymbol{x}) + \frac{1}{\sqrt{md}} \sum_{i=1}^{m} (\mathbf{u}_i - \mathbf{u}_{0,i})^T \boldsymbol{x} \mathbb{1}_{\{\mathbf{u}_{0,i}^T \boldsymbol{x} \geq 0\}} v_{0,i} + \frac{1}{\sqrt{m}} \sum_{i=1}^{m} (v_i - v_{0,i}) \sigma\left(\frac{1}{\sqrt{d}}\mathbf{u}_{0,i}^T\boldsymbol{x}\right)$$

$$+ \frac{1}{\sqrt{md}} \sum_{i=1}^{m} (\mathbf{u}_i - \mathbf{u}_{0,i})^T \boldsymbol{x} \mathbb{1}_{\{\mathbf{u}_{0,i}^T \boldsymbol{x} \geq 0\}} (v_i - v_{0,i}). \tag{13}$$

## B  DERIVATION OF DYNAMICS EQUATIONS

### B.1  SINGLE TRAINING EXAMPLE

The NQM can be equivalently written as:

$$g(\mathbf{u}, \mathbf{v}; \boldsymbol{x}) = g(\mathbf{u}_0, \mathbf{v}_0; \boldsymbol{x}) + \left\langle \mathbf{u} - \mathbf{u}_0, \nabla_{\mathbf{u}} g(\mathbf{u}, \mathbf{v}; \boldsymbol{x}) \Big|_{\mathbf{u}=\mathbf{u}_0, \mathbf{v}=\mathbf{v}_0} \right\rangle + \left\langle \mathbf{v} - \mathbf{v}_0, \nabla_{\mathbf{v}} g(\mathbf{u}, \mathbf{v}; \boldsymbol{x}) \Big|_{\mathbf{u}=\mathbf{u}_0, \mathbf{v}=\mathbf{v}_0} \right\rangle$$

$$+ \left\langle \mathbf{u} - \mathbf{u}_0, \frac{\partial^2 g(\mathbf{u}, \mathbf{v}; \boldsymbol{x})}{\partial \mathbf{u} \partial \mathbf{v}} \Big|_{\mathbf{u}=\mathbf{u}_0, \mathbf{v}=\mathbf{v}_0} (\mathbf{v} - \mathbf{v}_0) \right\rangle,$$

since $\frac{\partial^2 g}{\partial \mathbf{u}^2} = 0$ and $\frac{\partial^2 g}{\partial \mathbf{v}^2} = 0$.

And the tangent kernel $\lambda(\mathbf{u}, \mathbf{v}; \boldsymbol{x})$ takes the form

$$\lambda(\mathbf{u}, \mathbf{v}; \boldsymbol{x}) = \left\| \nabla_{\mathbf{u}} g(\mathbf{u}, \mathbf{v}; \boldsymbol{x}) \Big|_{\mathbf{u}=\mathbf{u}_0, \mathbf{v}=\mathbf{v}_0} + \frac{\partial^2 g(\mathbf{u}, \mathbf{v}; \boldsymbol{x})}{\partial \mathbf{u} \partial \mathbf{v}} \Big|_{\mathbf{u}=\mathbf{u}_0} (\mathbf{v} - \mathbf{v}_0) \right\|_F^2$$

$$+ \left\| \nabla_{\mathbf{v}} g(\mathbf{u}, \mathbf{v}; \boldsymbol{x}) \Big|_{\mathbf{u}=\mathbf{u}_0, \mathbf{v}=\mathbf{v}_0} + (\mathbf{u} - \mathbf{u}_0)^T \frac{\partial^2 g(\mathbf{u}, \mathbf{v}; \boldsymbol{x})}{\partial \mathbf{u} \partial \mathbf{v}} \Big|_{\mathbf{u}=\mathbf{u}_0, \mathbf{v}=\mathbf{v}_0} \right\|^2.$$

Here

$$\nabla_{\mathbf{u}_i} g(\mathbf{u}, \mathbf{v}; \boldsymbol{x}) \Big|_{\mathbf{u}=\mathbf{u}_0, \mathbf{v}=\mathbf{v}_0} = \frac{1}{\sqrt{md}} \sum_{i=1}^{m} v_{0,i} \mathbb{1}_{\{\mathbf{u}_{0,i}^T \boldsymbol{x} \geq 0\}} \boldsymbol{x}, \quad \forall i \in [m],$$

$$\nabla_{\mathbf{v}} g(\mathbf{u}, \mathbf{v}; \boldsymbol{x}) \Big|_{\mathbf{u}=\mathbf{u}_0, \mathbf{v}=\mathbf{v}_0} = \frac{1}{\sqrt{md}} \sigma\left(\mathbf{u}_0^T \boldsymbol{x}\right).$$

In the following, we will consider the dynamics of $g$ and $\lambda$ with GD, hence for simplicity of notations, we denote

$$\nabla_{\mathbf{u}} g(0) := \nabla_{\mathbf{u}} g(\mathbf{u}, \mathbf{v}; \boldsymbol{x})\Big|_{\mathbf{u}=\mathbf{u}_0, \mathbf{v}=\mathbf{v}_0},$$

$$\nabla_{\mathbf{v}} g(0) := \nabla_{\mathbf{v}} g(\mathbf{u}, \mathbf{v}; \boldsymbol{x})\Big|_{\mathbf{u}=\mathbf{u}_0, \mathbf{v}=\mathbf{v}_0},$$

$$\frac{\partial^2 g(0)}{\partial \mathbf{u} \partial \mathbf{v}} := \frac{\partial^2 g(\mathbf{u}, \mathbf{v}; \boldsymbol{x})}{\partial \mathbf{u} \partial \mathbf{v}}\Big|_{\mathbf{u}=\mathbf{u}_0, \mathbf{v}=\mathbf{v}_0}.$$

By gradient descent with learning rate $\eta$, at iteration $t$, we have the update equations for weights $\mathbf{u}$ and $\mathbf{v}$:

$$\mathbf{u}(t+1) = \mathbf{u}(t) - \eta(g(t) - y)\left(\nabla_{\mathbf{u}} g(0) + \frac{\partial^2 g(0)}{\partial \mathbf{u} \partial \mathbf{v}}(\mathbf{v}(t) - \mathbf{v}(0))\right),$$

$$\mathbf{v}(t+1) = \mathbf{v}(t) - \eta(g(t) - y)\left(\nabla_{\mathbf{v}} g(0) + (\mathbf{u}(t) - \mathbf{u}(0))^T \frac{\partial^2 g(0)}{\partial \mathbf{u} \partial \mathbf{v}}\right).$$

Then we plug them in the expression of $\lambda(t+1)$ and we get

$$\lambda(t+1) = \left\|\nabla_{\mathbf{u}} g(0) + \frac{\partial^2 g(0)}{\partial \mathbf{u} \partial \mathbf{v}}(\mathbf{v}(t+1) - \mathbf{v}(0))\right\|_F^2 + \left\|\nabla_{\mathbf{v}} g(0) + (\mathbf{u}(t+1) - \mathbf{u}(0))^T \frac{\partial^2 g(0)}{\partial \mathbf{u} \partial \mathbf{v}}\right\|^2$$

$$= \left\|\nabla_{\mathbf{u}} g(0) + \frac{\partial^2 g(0)}{\partial \mathbf{u} \partial \mathbf{v}}\left(\mathbf{v}(t) - \eta(g(t) - y)\left(\nabla_{\mathbf{v}} g(0) + (\mathbf{u}(t) - \mathbf{u}(0))^T \frac{\partial^2 g(0)}{\partial \mathbf{u} \partial \mathbf{v}}\right) - \mathbf{v}(0)\right)\right\|_F^2$$

$$+ \left\|\nabla_{\mathbf{v}} g(0) + \left(\mathbf{u}(t) - \eta(g(t) - y)\left(\nabla_{\mathbf{u}} g(0) + \frac{\partial^2 g(0)}{\partial \mathbf{u} \partial \mathbf{v}}(\mathbf{v}(t) - \mathbf{v}(0))\right) - \mathbf{u}(0)\right)^T \frac{\partial^2 g(0)}{\partial \mathbf{u} \partial \mathbf{v}}\right\|^2$$

$$= \lambda(t) + \eta^2(g(t) - y)^2 \left\|\frac{\partial^2 g(0)}{\partial \mathbf{u} \partial \mathbf{v}}\left(\nabla_{\mathbf{v}} g(0) + (\mathbf{u}(t) - \mathbf{u}(0))^T \frac{\partial^2 g(0)}{\partial \mathbf{u} \partial \mathbf{v}}\right)\right\|_F^2$$

$$+ \eta^2(g(t) - y)^2 \left\|\left(\nabla_{\mathbf{u}} g(0) + \frac{\partial^2 g(0)}{\partial \mathbf{u} \partial \mathbf{v}}(\mathbf{v}(t) - \mathbf{v}(0))\right)^T \frac{\partial^2 g(0)}{\partial \mathbf{u} \partial \mathbf{v}}\right\|^2$$

$$- 2\eta(g(t) - y)\left\langle \nabla_{\mathbf{u}} g(0) + \frac{\partial^2 g(0)}{\partial \mathbf{u} \partial \mathbf{v}}(\mathbf{v}(t) - \mathbf{v}(0)), \frac{\partial^2 g(0)}{\partial \mathbf{u} \partial \mathbf{v}}\left(\nabla_{\mathbf{v}} g(0) + (\mathbf{u}(t) - \mathbf{u}(0))^T \frac{\partial^2 g(0)}{\partial \mathbf{u} \partial \mathbf{v}}\right)\right\rangle$$

$$- 2\eta(g(t) - y)\left\langle \nabla_{\mathbf{v}} g(0) + (\mathbf{u}(t) - \mathbf{u}(0))^T \frac{\partial^2 g(0)}{\partial \mathbf{u} \partial \mathbf{v}}, \left(\nabla_{\mathbf{u}} g(0) + \frac{\partial^2 g(0)}{\partial \mathbf{u} \partial \mathbf{v}}(\mathbf{v}(t) - \mathbf{v}(0))\right)^T \frac{\partial^2 g(0)}{\partial \mathbf{u} \partial \mathbf{v}}\right\rangle.$$

Due to the structure of $\frac{\partial^2 g(0)}{\partial \mathbf{u} \partial \mathbf{v}}$, we have

$$\left\|\frac{\partial^2 g(0)}{\partial \mathbf{u} \partial \mathbf{v}}\left(\nabla_{\mathbf{v}} g(0) + (\mathbf{u}(t) - \mathbf{u}(0))^T \frac{\partial^2 g(0)}{\partial \mathbf{u} \partial \mathbf{v}}\right)\right\|_F^2 = \frac{\|\boldsymbol{x}\|^2}{md}\left\|\nabla_{\mathbf{v}} g(0) + (\mathbf{u}(t) - \mathbf{u}(0))^T \frac{\partial^2 g(0)}{\partial \mathbf{u} \partial \mathbf{v}}\right\|^2$$

$$= \frac{\|\boldsymbol{x}\|^2}{md}\|\nabla_{\mathbf{v}} g(t)\|^2,$$

and

$$\left\|\left(\nabla_{\mathbf{u}} g(0) + \frac{\partial^2 g(0)}{\partial \mathbf{u} \partial \mathbf{v}}(\mathbf{v}(t) - \mathbf{v}(0))\right)^T \frac{\partial^2 g(0)}{\partial \mathbf{u} \partial \mathbf{v}}\right\|^2 = \frac{\|\boldsymbol{x}\|^2}{md}\left\|\nabla_{\mathbf{u}} g(0) + \frac{\partial^2 g(0)}{\partial \mathbf{u} \partial \mathbf{v}}(\mathbf{v}(t) - \mathbf{v}(0))\right\|_F^2$$

$$= \frac{\|\boldsymbol{x}\|^2}{md}\|\nabla_{\mathbf{u}} g(t)\|_F^2.$$

Furthermore,

$$\left\langle \nabla_{\mathbf{u}} g(0) + \frac{\partial^2 g(0)}{\partial \mathbf{u} \partial \mathbf{v}} (\mathbf{v}(t) - \mathbf{v}(0)), \frac{\partial^2 g(0)}{\partial \mathbf{u} \partial \mathbf{v}} \left( \nabla_{\mathbf{v}} g(0) + (\mathbf{u}(t) - \mathbf{u}(0))^T \frac{\partial^2 g(0)}{\partial \mathbf{u} \partial \mathbf{v}} \right) \right\rangle$$

$$= \frac{\|\boldsymbol{x}\|^2}{md} \langle \mathbf{v}(t) - \mathbf{v}(0), \nabla_{\mathbf{v}} g(0) \rangle + \frac{\|\boldsymbol{x}\|^2}{md} \langle \nabla_{\mathbf{u}} g(0), \mathbf{u}(t) - \mathbf{u}(0) \rangle + \left\langle \nabla_{\mathbf{u}} g(0), \frac{\partial^2 g(0)}{\partial \mathbf{u} \partial \mathbf{v}} \nabla_{\mathbf{v}} g(0) \right\rangle$$

$$+ \left\langle \frac{\partial^2 g(0)}{\partial \mathbf{u} \partial \mathbf{v}} (\mathbf{v}(t) - \mathbf{v}(0)), \frac{\partial^2 g(0)}{\partial \mathbf{u} \partial \mathbf{v}} (\mathbf{u}(t) - \mathbf{u}(0))^T \frac{\partial^2 g(0)}{\partial \mathbf{u} \partial \mathbf{v}} \right\rangle$$

$$= \frac{\|\boldsymbol{x}\|^2}{md} \langle \mathbf{v}(t) - \mathbf{v}(0), \nabla_{\mathbf{v}} g(0) \rangle + \frac{\|\boldsymbol{x}\|^2}{md} \langle \nabla_{\mathbf{u}} g(0), \mathbf{u}(t) - \mathbf{u}(0) \rangle + g(0) + \frac{\|\boldsymbol{x}\|^2}{md} \left\langle \mathbf{v}(t) - \mathbf{v}(0), \frac{\partial^2 g(0)}{\partial \mathbf{u} \partial \mathbf{v}} (\mathbf{u}(t) - \mathbf{u}(0))^T \right\rangle$$

$$= g(t) \|\boldsymbol{x}\|^2 / md.$$

Similarly, we have

$$\left\langle \nabla_{\mathbf{v}} g(0) + (\mathbf{u}(t) - \mathbf{u}(0))^T \frac{\partial^2 g(0)}{\partial \mathbf{u} \partial \mathbf{v}}, \left( \nabla_{\mathbf{u}} g(0) + \frac{\partial^2 g(0)}{\partial \mathbf{u} \partial \mathbf{v}} (\mathbf{v}(t) - \mathbf{v}(0)) \right)^T \frac{\partial^2 g(0)}{\partial \mathbf{u} \partial \mathbf{v}} \right\rangle = g(t) \|\boldsymbol{x}\|^2 / md.$$

As a result,

$$\lambda(t+1) = \lambda(t) + \frac{\|\boldsymbol{x}\|^2}{md} \eta^2 (g(t) - y)^2 \lambda(t) - \frac{4 \|\boldsymbol{x}\|^2}{md} \eta (g(t) - y) g(t)$$

$$= \lambda(t) + \eta \frac{\|\boldsymbol{x}\|^2}{md} (g(t) - y)^2 \left( \eta \lambda(t) - 4 \frac{g(t)}{g(t) - y} \right).$$

For $g$, we plug the update equations for $\mathbf{u}$ and $\mathbf{v}$ in the expression of $g(t+1)$ and we can get

$$g(t+1) = g(0) + \langle \mathbf{u}(t+1) - \mathbf{u}(0), \nabla_{\mathbf{u}} g(0) \rangle + \langle \mathbf{v}(t+1) - \mathbf{v}(0), \nabla_{\mathbf{v}} g(0) \rangle$$

$$+ \left\langle \mathbf{u}(t+1) - \mathbf{u}(0), \frac{\partial^2 g(0)}{\partial \mathbf{u} \partial \mathbf{v}} (\mathbf{v}(t+1) - \mathbf{v}(0)) \right\rangle$$

$$= g(0) + \left\langle \mathbf{u}(t) - \eta(g(t) - y) \left( \nabla_{\mathbf{u}} g(0) + \frac{\partial^2 g(0)}{\partial \mathbf{u} \partial \mathbf{v}} (\mathbf{v}(t) - \mathbf{v}(0)) \right) - \mathbf{u}(0), \nabla_{\mathbf{u}} g(0) \right\rangle$$

$$+ \left\langle \mathbf{v}(t) - \eta(g(t) - y) \left( \nabla_{\mathbf{v}} g(0) + (\mathbf{u}(t) - \mathbf{u}(0))^T \frac{\partial^2 g(0)}{\partial \mathbf{u} \partial \mathbf{v}} \right) - \mathbf{v}(0), \nabla_{\mathbf{v}} g(0) \right\rangle$$

$$+ \left\langle \mathbf{u}(t) - \eta(g(t) - y) \left( \nabla_{\mathbf{u}} g(0) + \frac{\partial^2 g(0)}{\partial \mathbf{u} \partial \mathbf{v}} (\mathbf{v}(t) - \mathbf{v}(0)) \right) - \mathbf{u}(0), \right.$$

$$\left. \frac{\partial^2 g(0)}{\partial \mathbf{u} \partial \mathbf{v}} \left( \mathbf{v}(t) - \eta(g(t) - y) \left( \nabla_{\mathbf{v}} g(0) + (\mathbf{u}(t) - \mathbf{u}(0))^T \frac{\partial^2 g(0)}{\partial \mathbf{u} \partial \mathbf{v}} \right) - \mathbf{v}(0) \right) \right\rangle$$

$$= g(t) - \eta(g(t) - y) \left\langle \nabla_{\mathbf{u}} g(0) + \frac{\partial^2 g(0)}{\partial \mathbf{u} \partial \mathbf{v}} (\mathbf{v}(t) - \mathbf{v}(0)), \nabla_{\mathbf{u}} g(0) \right\rangle$$

$$- \eta(g(t) - y) \left\langle \nabla_{\mathbf{v}} g(0) + (\mathbf{u}(t) - \mathbf{u}(0))^T \frac{\partial^2 g(0)}{\partial \mathbf{u} \partial \mathbf{v}}, \nabla_{\mathbf{v}} g(0) \right\rangle$$

$$+ \eta^2 (g(t) - y)^2 \left\langle \nabla_{\mathbf{u}} g(0) + \frac{\partial^2 g(0)}{\partial \mathbf{u} \partial \mathbf{v}} (\mathbf{v}(t) - \mathbf{v}(0)), \frac{\partial^2 g(0)}{\partial \mathbf{u} \partial \mathbf{v}} \left( \nabla_{\mathbf{v}} g(0) + (\mathbf{u}(t) - \mathbf{u}(0))^T \frac{\partial^2 g(0)}{\partial \mathbf{u} \partial \mathbf{v}} \right) \right\rangle$$

$$- \eta(g(t) - y) \left\langle \mathbf{u}(t) - \mathbf{u}(0), \frac{\partial^2 g(0)}{\partial \mathbf{u} \partial \mathbf{v}} \left( \nabla_{\mathbf{v}} g(0) + (\mathbf{u}(t) - \mathbf{u}(0))^T \frac{\partial^2 g(0)}{\partial \mathbf{u} \partial \mathbf{v}} \right) \right\rangle$$

$$- \eta(g(t) - y) \left\langle \nabla_{\mathbf{u}} g(0) + \frac{\partial^2 g(0)}{\partial \mathbf{u} \partial \mathbf{v}} (\mathbf{v}(t) - \mathbf{v}(0)), \frac{\partial^2 g(0)}{\partial \mathbf{u} \partial \mathbf{v}} (\mathbf{v}(t) - \mathbf{v}(0)) \right\rangle$$

$$= g(t) - \eta(g(t) - y) \lambda(t)$$

$$+ \eta^2 (g(t) - y)^2 \left\langle \nabla_{\mathbf{u}} g(0) + \frac{\partial^2 g(0)}{\partial \mathbf{u} \partial \mathbf{v}} (\mathbf{v}(t) - \mathbf{v}(0)), \frac{\partial^2 g(0)}{\partial \mathbf{u} \partial \mathbf{v}} \left( \nabla_{\mathbf{v}} g(0) + (\mathbf{u}(t) - \mathbf{u}(0))^T \frac{\partial^2 g(0)}{\partial \mathbf{u} \partial \mathbf{v}} \right) \right\rangle$$

$$= g(t) - \eta(g(t) - y) \lambda(t) + \frac{\|\boldsymbol{x}\|^2}{md} \eta^2 (g(t) - y)^2 g(t)$$

Therefore,

$$g(t+1) - y = \left(1 - \eta\lambda(t) + \frac{\|\boldsymbol{x}\|^2}{md}\eta^2(g(t)-y)g(t)\right)(g(t)-y).$$

## B.2 MULTIPLE TRAINING EXAMPLES

We follow the similar notation on the first and second order derivative of $g$ with Appendix B.1. Specifically, for $k \in [n]$, we denote

$$\nabla_{\mathbf{u}}g_k(0) := \nabla_{\mathbf{u}}g(\mathbf{u}, \mathbf{v}; x_k)\Big|_{\mathbf{u}=\mathbf{u}_0, \mathbf{v}=\mathbf{v}_0},$$

$$\nabla_{\mathbf{v}}g_k(0) := \nabla_{\mathbf{v}}g(\mathbf{u}, \mathbf{v}; x_k)\Big|_{\mathbf{u}=\mathbf{u}_0, \mathbf{v}=\mathbf{v}_0},$$

$$\frac{\partial^2 g_k(0)}{\partial\mathbf{u}\partial\mathbf{v}} := \frac{\partial^2 g(\mathbf{u}, \mathbf{v}; x_k)}{\partial\mathbf{u}\partial\mathbf{v}}\Big|_{\mathbf{u}=\mathbf{u}_0, \mathbf{v}=\mathbf{v}_0}.$$

By GD with learning rate $\eta$, we have the update equations for weights $\mathbf{u}$ and $\mathbf{v}$ at iteration $t$:

$$\mathbf{u}(t+1) = \mathbf{u}(t) - \eta\sum_{k=1}^{n}(g_k(t) - y_k)\left(\nabla_{\mathbf{u}}g_k(0) + \frac{\partial^2 g_k(0)}{\partial\mathbf{u}\partial\mathbf{v}}(\mathbf{v}(t) - \mathbf{v}(0))\right),$$

$$\mathbf{v}(t+1) = \mathbf{v}(t) - \eta\sum_{k=1}^{n}(g_k(t) - y_k)\left(\nabla_{\mathbf{v}}g_k(0) + (\mathbf{u}(t) - \mathbf{u}(0))^T\frac{\partial^2 g_k(0)}{\partial\mathbf{u}\partial\mathbf{v}}\right).$$

We consider the evolution of $K(t)$ first.

$$K_{i,j}(t+1) = \left\langle \nabla_{\mathbf{u}}g_i(0) + \frac{\partial^2 g_i(0)}{\partial\mathbf{u}\partial\mathbf{v}}(\mathbf{v}(t+1) - \mathbf{v}(0)), \nabla_{\mathbf{u}}g_j(0) + \frac{\partial^2 g_j(0)}{\partial\mathbf{u}\partial\mathbf{v}}(\mathbf{v}(t+1) - \mathbf{v}(0)) \right\rangle$$

$$+ \left\langle \nabla_{\mathbf{v}}g_i(0) + (\mathbf{u}(t+1) - \mathbf{u}(0))^T\frac{\partial^2 g_i(0)}{\partial\mathbf{u}\partial\mathbf{v}}, \nabla_{\mathbf{v}}g_j(0) + (\mathbf{u}(t+1) - \mathbf{u}(0))^T\frac{\partial^2 g_j(0)}{\partial\mathbf{u}\partial\mathbf{v}} \right\rangle$$

$$= K_{i,j}(t) - \left\langle \eta\frac{\partial^2 g_i(0)}{\partial\mathbf{u}\partial\mathbf{v}}\sum_{k=1}^{n}(g_k(t) - y_k)\left(\nabla_{\mathbf{v}}g_k(0) + (\mathbf{u}(t) - \mathbf{u}(0))^T\frac{\partial^2 g_k(0)}{\partial\mathbf{u}\partial\mathbf{v}}\right), \right.$$

$$\left. \nabla_{\mathbf{u}}g_j(0) + \frac{\partial^2 g_j(0)}{\partial\mathbf{u}\partial\mathbf{v}}(\mathbf{v}(t) - \mathbf{v}(0)) \right\rangle$$

$$- \left\langle \eta\frac{\partial^2 g_j(0)}{\partial\mathbf{u}\partial\mathbf{v}}\sum_{k=1}^{n}(g_k(t) - y_k)\left(\nabla_{\mathbf{v}}g_k(0) + (\mathbf{u}(t) - \mathbf{u}(0))^T\frac{\partial^2 g_k(0)}{\partial\mathbf{u}\partial\mathbf{v}}\right), \nabla_{\mathbf{u}}g_i(0) + \frac{\partial^2 g_i(0)}{\partial\mathbf{u}\partial\mathbf{v}}(\mathbf{v}(t) - \mathbf{v}(0)) \right\rangle$$

$$+ \left\langle \eta\frac{\partial^2 g_i(0)}{\partial\mathbf{u}\partial\mathbf{v}}\sum_{k=1}^{n}(g_k(t) - y_k)\left(\nabla_{\mathbf{v}}g_k(0) + (\mathbf{u}(t) - \mathbf{u}(0))^T\frac{\partial^2 g_k(0)}{\partial\mathbf{u}\partial\mathbf{v}}\right), \right.$$

$$\left. \eta\frac{\partial^2 g_j(0)}{\partial\mathbf{u}\partial\mathbf{v}}\sum_{k=1}^{n}(g_k(t) - y_k)\left(\nabla_{\mathbf{v}}g_k(0) + (\mathbf{u}(t) - \mathbf{u}(0))^T\frac{\partial^2 g_k(0)}{\partial\mathbf{u}\partial\mathbf{v}}\right) \right\rangle$$

$$- \left\langle \eta\frac{\partial^2 g_j(0)}{\partial\mathbf{u}\partial\mathbf{v}}\sum_{k=1}^{n}(g_k(t) - y_k)\left(\nabla_{\mathbf{u}}g_k(0) + \frac{\partial^2 g_k(0)}{\partial\mathbf{u}\partial\mathbf{v}}(\mathbf{v}(t) - \mathbf{v}(0))\right), \nabla_{\mathbf{v}}g_i(0) + (\mathbf{u}(t) - \mathbf{u}(0))^T\frac{\partial^2 g_i(0)}{\partial\mathbf{u}\partial\mathbf{v}} \right\rangle$$

$$- \left\langle \eta\frac{\partial^2 g_i(0)}{\partial\mathbf{u}\partial\mathbf{v}}\sum_{k=1}^{n}(g_k(t) - y_k)\left(\nabla_{\mathbf{u}}g_k(0) + \frac{\partial^2 g_k(0)}{\partial\mathbf{u}\partial\mathbf{v}}(\mathbf{v}(t) - \mathbf{v}(0))\right), \nabla_{\mathbf{v}}g_j(0) + (\mathbf{u}(t) - \mathbf{u}(0))^T\frac{\partial^2 g_j(0)}{\partial\mathbf{u}\partial\mathbf{v}} \right\rangle$$

$$+ \left\langle \eta\frac{\partial^2 g_i(0)}{\partial\mathbf{u}\partial\mathbf{v}}\sum_{k=1}^{n}(g_k(t) - y_k)\left(\nabla_{\mathbf{u}}g_k(0) + \frac{\partial^2 g_k(0)}{\partial\mathbf{u}\partial\mathbf{v}}(\mathbf{v}(t) - \mathbf{v}(0))\right), \right.$$

$$\left. \eta\frac{\partial^2 g_j(0)}{\partial\mathbf{u}\partial\mathbf{v}}\sum_{k=1}^{n}(g_k(t) - y_k)\left(\nabla_{\mathbf{u}}g_k(0) + \frac{\partial^2 g_k(0)}{\partial\mathbf{u}\partial\mathbf{v}}(\mathbf{v}(t) - \mathbf{v}(0))\right) \right\rangle.$$

We separate the data into two sets according to their sign:
$$\mathcal{S}_+ := \{i : x_i \geq 0, i \in [n]\}, \quad \mathcal{S}_- := \{i : x_i < 0, i \in [n]\}.$$

We consider two scenarios: (1) $x_i$ and $x_j$ have different signs; (2) $x_i$ and $x_j$ have the same sign.

**(1)** With simple calculation, we get if $x_i$ and $x_j$ have different signs, i.e., $i \in \mathcal{S}_+, j \in \mathcal{S}_-$ or $i \in \mathcal{S}_-, j \in \mathcal{S}_+$,
$$\frac{\partial^2 g_i(0)}{\partial \mathbf{u} \partial \mathbf{v}} \frac{\partial^2 g_j(0)}{\partial \mathbf{u} \partial \mathbf{v}} = 0, \quad \frac{\partial^2 g_i(0)}{\partial \mathbf{u} \partial \mathbf{v}} \nabla_\mathbf{u} g_j(0) = 0, \quad \frac{\partial^2 g_i(0)}{\partial \mathbf{u} \partial \mathbf{v}} \nabla_\mathbf{v} g_j(0) = 0.$$

Without lose of generality, we assume $i \in \mathcal{S}_+, j \in \mathcal{S}_-$. Then we have
$$K_{i,j}(t+1) = K_{i,j}(t).$$

**(2)** If $x_i$ and $x_j$ have the same sign, i.e., $i, j \in \mathcal{S}_+$ or $i, j \in \mathcal{S}_-$,
$$\frac{\partial^2 g_i(0)}{\partial \mathbf{u} \partial \mathbf{v}} \frac{\partial^2 g_j(0)}{\partial \mathbf{u} \partial \mathbf{v}} = \frac{1}{\sqrt{m}} \frac{\partial^2 g_i(0)}{\partial \mathbf{u} \partial \mathbf{v}} x_j, \quad \frac{\partial^2 g_i(0)}{\partial \mathbf{u} \partial \mathbf{v}} \nabla_\mathbf{u} g_j(0) = \frac{1}{\sqrt{m}} \nabla_\mathbf{u} g_i(0) x_j, \quad \frac{\partial^2 g_i(0)}{\partial \mathbf{u} \partial \mathbf{v}} \nabla_\mathbf{v} g_j(0) = \frac{1}{\sqrt{m}} \nabla_\mathbf{v} g_i(0) x_j.$$

For $i, j \in \mathcal{S}_+$, we have
$$K_{i,j}(t+1) = K_{i,j}(t) - \frac{2\eta}{\sqrt{m}} \sum_{k \in \mathcal{S}_+} (g_k(t) - y_k) x_i \left\langle \nabla_\mathbf{v} g_k(0) + (\mathbf{u}(t) - \mathbf{u}(0))^T \frac{\partial^2 g_k(0)}{\partial \mathbf{u} \partial \mathbf{v}} \right.,$$

$$\left. \nabla_\mathbf{u} g_j(0) + \frac{\partial^2 g_j(0)}{\partial \mathbf{u} \partial \mathbf{v}} (\mathbf{v}(t) - \mathbf{v}(0)) \right\rangle$$

$$- \frac{2\eta}{\sqrt{m}} \sum_{k \in \mathcal{S}_+} (g_k(t) - y_k) x_i \left\langle \nabla_\mathbf{u} g_k(0) + \frac{\partial^2 g_k(0)}{\partial \mathbf{u} \partial \mathbf{v}} (\mathbf{v}(t) - \mathbf{v}(0)), \nabla_\mathbf{v} g_j(0) + (\mathbf{u}(t) - \mathbf{u}(0))^T \frac{\partial^2 g_j(0)}{\partial \mathbf{u} \partial \mathbf{v}} \right\rangle$$

$$+ \frac{\eta^2}{m} x_i x_j \left\| \sum_{k \in \mathcal{S}_+} (g_k(t) - y_k) \left( \nabla_\mathbf{v} g_k(0) + (\mathbf{u}(t) - \mathbf{u}(0))^T \frac{\partial^2 g_k(0)}{\partial \mathbf{u} \partial \mathbf{v}} \right) \right\|^2$$

$$+ \frac{\eta^2}{m} x_i x_j \left\| \sum_{k \in \mathcal{S}_+} (g_k(t) - y_k) \left( \nabla_\mathbf{u} g_k(0) + \frac{\partial^2 g_k(0)}{\partial \mathbf{u} \partial \mathbf{v}} (\mathbf{v}(t) - \mathbf{v}(0)) \right) \right\|^2$$

$$= K_{i,j}(t) - \frac{4\eta}{m} x_i x_j \sum_{k \in \mathcal{S}_+} (g_k(t) - y_k) g_k(t) + \frac{\eta^2}{m} x_i x_j \left( (\mathbf{g}(t) - \mathbf{y}) \odot \boldsymbol{m}_+ \right)^T K(t) \left( (\mathbf{g}(t) - \mathbf{y}) \odot \boldsymbol{m}_+ \right)$$

$$= K_{i,j}(t) - \frac{4\eta}{m} x_i x_j \left( (\mathbf{g}(t) - \mathbf{y}) \odot \boldsymbol{m}_+ \right)^T \left( \mathbf{g}(t) \odot \boldsymbol{m}_+ \right)$$

$$+ \frac{\eta^2}{m} x_i x_j \left( (\mathbf{g}(t) - \mathbf{y}) \odot \boldsymbol{m}_+ \right)^T K(t) \left( (\mathbf{g}(t) - \mathbf{y}) \odot \boldsymbol{m}_+ \right).$$

Similarly, for $i, j \in \mathcal{S}_-$, we have
$$K_{i,j}(t+1) = K_{i,j}(t) - \frac{4\eta}{m} x_i x_j \left( (\mathbf{g}(t) - \mathbf{y}) \odot \boldsymbol{m}_- \right)^T \left( \mathbf{g}(t) \odot \boldsymbol{m}_- \right)$$

$$+ \frac{\eta^2}{m} x_i x_j \left( (\mathbf{g}(t) - \mathbf{y}) \odot \boldsymbol{m}_- \right)^T K(t) \left( (\mathbf{g}(t) - \mathbf{y}) \odot \boldsymbol{m}_- \right).$$

Combining the results together, we have
$$K(t+1) = K(t) + \frac{\eta^2}{m} \left( (\mathbf{g}(t) - \mathbf{y}) \odot \boldsymbol{m}_+ \right)^T K(t) \left( (\mathbf{g}(t) - \mathbf{y}) \odot \boldsymbol{m}_+ \right) \boldsymbol{p}_1 \boldsymbol{p}_1^T$$

$$+ \frac{\eta^2}{m} \left( (\mathbf{g}(t) - \mathbf{y}) \odot \boldsymbol{m}_- \right)^T K(t) \left( (\mathbf{g}(t) - \mathbf{y}) \odot \boldsymbol{m}_- \right) \boldsymbol{p}_2 \boldsymbol{p}_2^T$$

$$- \frac{4\eta}{m} \left( (\mathbf{g}(t) - \mathbf{y}) \odot \boldsymbol{m}_+ \right)^T \left( \mathbf{g}(t) \odot \boldsymbol{m}_+ \right) \boldsymbol{p}_1 \boldsymbol{p}_1^T$$

$$- \frac{4\eta}{m} \left( (\mathbf{g}(t) - \mathbf{y}) \odot \boldsymbol{m}_- \right)^T \left( \mathbf{g}(t) \odot \boldsymbol{m}_- \right) \boldsymbol{p}_2 \boldsymbol{p}_2^T.$$

Now we derive the evolution of $\mathbf{g}(t) - \mathbf{y}$. Suppose $i \in \mathcal{S}_+$. Then we have

$$
\begin{aligned}
g_i(t+1) &= g_i(0) + \langle \mathbf{u}(t+1) - \mathbf{u}(0), \nabla_{\mathbf{u}} g_i(0) \rangle + \langle \mathbf{v}(t+1) - \mathbf{v}(0), \nabla_{\mathbf{v}} g_i(0) \rangle \\
&\quad + \left\langle \mathbf{u}(t+1) - \mathbf{u}(0), \frac{\partial^2 g_i(0)}{\partial \mathbf{u} \partial \mathbf{v}} (\mathbf{v}(t+1) - \mathbf{v}(0)) \right\rangle \\
&= g_i(t) - \eta \left\langle \sum_{k=1}^{n} (g_k(t) - y_k) \left( \nabla_{\mathbf{u}} g_k(0) + \frac{\partial^2 g_k(0)}{\partial \mathbf{u} \partial \mathbf{v}} (\mathbf{v}(t) - \mathbf{v}(0)) \right), \nabla_{\mathbf{u}} g_i(0) \right\rangle \\
&\quad - \eta \left\langle \sum_{k=1}^{n} (g_k(t) - y_k) \left( \nabla_{\mathbf{v}} g_k(0) + (\mathbf{u}(t) - \mathbf{u}(0))^T \frac{\partial^2 g_k(0)}{\partial \mathbf{u} \partial \mathbf{v}} \right), \nabla_{\mathbf{v}} g_i(0) \right\rangle \\
&\quad - \eta \left\langle \sum_{k=1}^{n} (g_k(t) - y_k) \left( \nabla_{\mathbf{u}} g_k(0) + \frac{\partial^2 g_k(0)}{\partial \mathbf{u} \partial \mathbf{v}} (\mathbf{v}(t) - \mathbf{v}(0)) \right), \frac{\partial^2 g_i(0)}{\partial \mathbf{u} \partial \mathbf{v}} (\mathbf{v}(t) - \mathbf{v}(0)) \right\rangle \\
&\quad - \eta \left\langle \sum_{k=1}^{n} (g_k(t) - y_k) \left( \nabla_{\mathbf{v}} g_k(0) + (\mathbf{u}(t) - \mathbf{u}(0))^T \frac{\partial^2 g_k(0)}{\partial \mathbf{u} \partial \mathbf{v}} \right), (\mathbf{u}(t) - \mathbf{u}(0)^T \frac{\partial^2 g_i(0)}{\partial \mathbf{u} \partial \mathbf{v}} \right\rangle \\
&\quad + \eta^2 \left\langle \sum_{k=1}^{n} (g_k(t) - y_k) \left( \nabla_{\mathbf{u}} g_k(0) + \frac{\partial^2 g_k(0)}{\partial \mathbf{u} \partial \mathbf{v}} (\mathbf{v}(t) - \mathbf{v}(0)) \right), \right. \\
&\qquad\qquad \left. \frac{\partial^2 g_i(0)}{\partial \mathbf{u} \partial \mathbf{v}} \sum_{k=1}^{n} (g_k(t) - y_k) \left( \nabla_{\mathbf{v}} g_k(0) + (\mathbf{u}(t) - \mathbf{u}(0))^T \frac{\partial^2 g_k(0)}{\partial \mathbf{u} \partial \mathbf{v}} \right) \right\rangle \\
&= g_i(t) - \eta \sum_{k \in \mathcal{S}_+} (g_k(t) - y_k) K_{k,i}(t) + \frac{\eta^2}{m} \sum_{k \in \mathcal{S}_+} \sum_{j \in \mathcal{S}_+} (g_k(t) - y_k)(g_j(t) - y_j) g_j(t) x_k x_i.
\end{aligned}
$$

Similarly, for $i \in \mathcal{S}_-$, we have

$$
g_i(t+1) = g_i(t) - \eta \sum_{k \in \mathcal{S}_-} (g_k(t) - y_k) K_{k,i}(t) + \frac{\eta^2}{m} \sum_{k \in \mathcal{S}_- f} \sum_{j \in \mathcal{S}_-} (g_k(t) - y_k)(g_j(t) - y_j) g_j(t) x_k x_i.
$$

Combining the results together, we have

$$
\begin{aligned}
\mathbf{g}(t+1) - \mathbf{y} = \Bigg( & I - \eta K(t) + \frac{\eta^2}{m} ((\mathbf{g}(t) - \mathbf{y}) \odot \boldsymbol{m}_+)^T (\mathbf{g}(t) \odot \boldsymbol{m}_+) \boldsymbol{p}_1 \boldsymbol{p}_1^T \\
& + \frac{\eta^2}{m} ((\mathbf{g}(t) - \mathbf{y}) \odot \boldsymbol{m}_-)^T (\mathbf{g}(t) \odot \boldsymbol{m}_-) \boldsymbol{p}_2 \boldsymbol{p}_2^T \Bigg) (\mathbf{g}(t) - \mathbf{y}).
\end{aligned}
$$

## C  PROOF OF PROPOSITION 1

**Restate Proposition 1:** *For any $\mathbf{u}, \mathbf{v} \in \mathbb{R}^m$, $\mathrm{rank}(K) \leq 2$. Furthermore, $\boldsymbol{p}_1, \boldsymbol{p}_2$ are eigenvectors of $K$, where $p_{1,i} = x_i \mathbb{1}_{\{i \in \mathcal{S}_+\}}$, $p_{2,i} = x_i \mathbb{1}_{\{i \in \mathcal{S}_-\}}$, for $i \in [n]$.*

*Proof.* By Definition 1,

$$
K_{i,j} = \frac{1}{m} \sum_{k=1}^{m} (v_k^2 + u_k^2) x_i x_j \mathbb{1}_{\{u_k x_i \geq 0\}} \mathbb{1}_{\{u_k x_j \geq 0\}}, \quad i, j \in [n].
$$

By definition of eigenvector, we can see

$$\sum_{j=1}^{n} K_{i,j} p_{1,j} = \frac{1}{m} \sum_{j=1}^{n} \sum_{k=1}^{m} (v_k^2 + u_k^2) x_i x_j^2 \mathbb{1}_{\{u_k x_i \geq 0\}} \mathbb{1}_{\{u_k x_j \geq 0\}} \mathbb{1}_{\{j \in \mathcal{S}_+\}}$$

$$= \sum_{j=1}^{n} x_j^2 \mathbb{1}_{\{j \in \mathcal{S}_+\}} \frac{1}{m} \sum_{k=1}^{m} (v_k^2 + u_k^2) x_i \mathbb{1}_{\{u_k x_i \geq 0\}} \mathbb{1}_{\{u_k x_j \geq 0\}}$$

$$= x_i \mathbb{1}_{\{x_i \in S_+\}} \sum_{j=1}^{n} x_j^2 \mathbb{1}_{\{j \in \mathcal{S}_+\}} \frac{1}{m} \sum_{k=1}^{m} (v_k^2 + u_k^2) \mathbb{1}_{\{u_k x_j \geq 0\}},$$

where we use the fact that if $x_i x_j < 0$, $K_{i,j} = 0$.

As $p_{1,i} = x_i \mathbb{1}_{\{x_i \in \mathcal{S}_+\}}$ and $\sum_{j=1}^{n} x_j^2 \mathbb{1}_{\{j \in \mathcal{S}_+\}} \frac{1}{m} \sum_{k=1}^{m} (v_k^2 + u_k^2) \mathbb{1}_{\{u_k x_j \geq 0\}}$ does not depend on $i$, we can see $p_1$ is an eigenvector of $K$ with corresponding eigenvalue $\lambda_1 = \sum_{j=1}^{n} x_j^2 \mathbb{1}_{\{j \in \mathcal{S}_+\}} \frac{1}{m} \sum_{k=1}^{m} (v_k^2 + u_k^2) \mathbb{1}_{\{u_k x_j \geq 0\}}$.

The same analysis can be applied to show $p_2$ is another eigenvector of $K$ with corresponding $\lambda_2 = \sum_{j=1}^{n} x_j^2 \mathbb{1}_{\{j \in \mathcal{S}_-\}} \frac{1}{m} \sum_{k=1}^{m} (v_k^2 + u_k^2) \mathbb{1}_{\{u_k x_j \geq 0\}}$.

For the rank of $K$, it is not hard to verify that $K = \lambda_1 p_1 p_1^T + \lambda_2 p_2 p_2^T$ hence the rank of $K$ is at most 2. $\qquad \square$

## D  SCALE OF THE TANGENT KERNEL FOR SINGLE TRAINING EXAMPLE

**Proposition 2** (Scale of tangent kernel). *For any $\delta \in (0,1)$, if $m \geq c' \log(4/\delta)$ where $c'$ is an absolute constant, with probability at least $1 - \delta$, $\|\boldsymbol{x}\|^2/(2d) \leq \lambda(0) \leq 3\|\boldsymbol{x}\|^2/(2d)$.*

*Proof.* Note that when $t = 0$,

$$\lambda(0) = \frac{1}{md} \sum_{i=1}^{m} \left( \mathbf{u}_{0,i}^T \boldsymbol{x} \mathbb{1}_{\{\mathbf{u}_{0,i}^T \boldsymbol{x} \geq 0\}} \right)^2 + \frac{1}{md} \sum_{i=1}^{m} (v_{0,i})^2 \|\boldsymbol{x}\|^2 \left( \mathbb{1}_{\{\mathbf{u}_{0,i}^T \boldsymbol{x} \geq 0\}} \right)^2.$$

According to NTK initialization, for each $i \in [m]$, $v_{0,i} \sim \mathcal{N}(0,1)$ and $\mathbf{u}_{0,i} \sim \mathcal{N}(0,I)$. We consider the random variable

$$\zeta_i := \mathbf{u}_{0,i}^T \boldsymbol{x} \mathbb{1}_{\{\mathbf{u}_{0,i}^T \boldsymbol{x} \geq 0\}}, \quad \xi_i := v_{0,i} \mathbb{1}_{\{\mathbf{u}_{0,i}^T \boldsymbol{x} \geq 0\}}.$$

it is not hard to see that $\zeta_i$ and $\xi_i$ are sub-guassian since $\mathbf{u}_{0,i}^T \boldsymbol{x}$ and $v_{0,i}$ are sub-gaussian. Specifically, for any $t \geq 0$,

$$\mathbb{P}\{|\zeta_i| \geq t\} \leq \mathbb{P}\{|\mathbf{u}_{0,i}^T \boldsymbol{x}| \geq t\} \leq 2 \exp\left(-t^2/(2\|\boldsymbol{x}\|^2)\right),$$

$$\mathbb{P}\{|\xi_i| \geq t\} \leq \mathbb{P}\{|v_{0,i}| \geq t\} \leq 2 \exp\left(-t^2/2\right),$$

where the second inequality comes from the definition of sub-gaussian variables.

Since $\xi_i$ is sub-gaussian, by definition, $\xi^2$ is sub-exponential, and its sub-exponential norm is bounded:

$$\|\xi_i^2\|_{\psi_1} \leq \|\xi_i\|_{\psi_2}^2 \leq C,$$

where $C > 0$ is a absolute constant. Similarly we have $\|\zeta_i\|_{\psi_2}^2 \leq C\|\boldsymbol{x}\|^2$.

By Bernstein's inequality, for every $t \geq 0$, we have

$$\mathbb{P}\left\{ \left| \sum_{i=1}^{m} \xi_i^2 - \frac{m}{2} \right| \geq t \right\} \leq 2 \exp\left( -c \min\left( \frac{t^2}{\sum_{i=1}^{m} \|\xi_i^2\|_{\psi_1}^2}, \frac{t}{\max_i \|\xi_i^2\|_{\psi_1}} \right) \right),$$

where $c > 0$ is an absolute constant.

Letting $t = m/4$, we have with probability at least $1 - 2 \exp\left(-m/c'\right)$,

$$\frac{m}{4} \leq \sum_{i=1}^{m} \xi_i^2 \leq \frac{3m}{4},$$

where $c' = c/(4C)$.

Similarity, we have with probability at least $1 - 2\exp\left(-m/c'\right)$,

$$\frac{m}{4}\|\boldsymbol{x}\|^2 \leq \sum_{i=1}^{m} \zeta_i^2 \leq \frac{3m}{4}\|\boldsymbol{x}\|^2.$$

As a result, using union bound, we have probability at least $1 - 4\exp\left(-m/c'\right)$,

$$\frac{\|\boldsymbol{x}\|^2}{2d} \leq \lambda(0) \leq \frac{3\|\boldsymbol{x}\|^2}{2d}.$$

$\square$

# E   SCALE OF THE TANGENT KERNEL FOR MULTIPLE TRAINING EXAMPLES

*Proof.* As shown in Proposition 1, $\boldsymbol{p}_1$ and $\boldsymbol{p}_2$ are eigenvectors of $K$, hence we have two eigenvalues:

$$\lambda_1(0) = \frac{\boldsymbol{p}_1^T K(0)\boldsymbol{p}_1}{\|\boldsymbol{p}_1\|^2}, \quad \lambda_2(0) = \frac{\boldsymbol{p}_2^T K(0)\boldsymbol{p}_2}{\|\boldsymbol{p}_2\|^2}.$$

Take $\lambda_1(0)$ as an example:

$$\lambda_1(0)\|\boldsymbol{p}_1\|^2 = \sum_{i,j=1}^{n} x_i x_j \mathbb{1}_{\{x_i \geq 0\}} \mathbb{1}_{\{x_j \geq 0\}} \sum_{k=1}^{m} (u_{0,k}^2 + v_{0,k}^2) x_i x_j \mathbb{1}_{\{u_{0,k} x_i \geq 0\}} \mathbb{1}_{\{u_{0,k} x_j \geq 0\}}$$

$$= \sum_{k=1}^{m} (u_{0,k}^2 + v_{0,k}^2) \left(\mathbb{1}_{\{u_{0,k} \geq 0\}}\right)^2 \sum_{i,j=1}^{n} x_i^2 x_j^2 \mathbb{1}_{\{x_i \geq 0\}} \mathbb{1}_{\{x_j \geq 0\}}.$$

Similar to the proof of Proposition 2, we consider $\xi_k := v_{0,k} \mathbb{1}_{\{u_{0,k} \geq 0\}}$ which is a sub-gaussian random variable. Hence $\xi_k^2$ is sub-exponential so that $\|\xi_k^2\|_{\psi_1} \leq C$ where $C > 0$ is an absolute constant. By Bernstein's inequality, for every $t \geq 0$, we have

$$\mathbb{P}\left\{\left|\sum_{i=1}^{m} \xi_i^2 - \frac{m}{2}\right| \geq t\right\} \leq 2\exp\left(-c\min\left(\frac{t^2}{\sum_{i=1}^{m}\|\xi_i^2\|_{\psi_1}^2}, \frac{t}{\max_i \|\xi_i^2\|_{\psi_1}}\right)\right),$$

where $c > 0$ is an absolute constant.

Letting $t = m/4$, we have with probability at least $1 - 2\exp\left(-m/c'\right)$,

$$\frac{m}{4} \leq \sum_{i=1}^{m} \xi_i^2 \leq \frac{3m}{4},$$

where $c' = c/(4C)$.

The same analysis applies to $\zeta_k := u_{0,k} \mathbb{1}_{\{u_{0,k} \geq 0\}}$ as well and we have with probability at least $1 - 2\exp\left(-m/c'\right)$,

$$\frac{m}{4} \leq \sum_{i=1}^{m} \zeta_i^2 \leq \frac{3m}{4}.$$

As a result, we have probability at least $1 - 4\exp\left(-m/c'\right)$,

$$\lambda_1(0)\|\boldsymbol{p}_1\|^2 = \frac{1}{m} \sum_{i=k}^{m} (u_{0,k}^2 + v_{0,k}^2) \left(\mathbb{1}_{\{u_k(0) \geq 0\}}\right)^2 \sum_{i,j=1}^{n} x_i^2 x_j^2 \mathbb{1}_{\{x_i \geq 0\}} \mathbb{1}_{\{x_j \geq 0\}}$$

$$\in \left[\frac{1}{2} \sum_{i,j=1}^{n} x_i^2 x_j^2 \mathbb{1}_{\{x_i \geq 0\}} \mathbb{1}_{\{x_j \geq 0\}}, \frac{3}{2} \sum_{i,j=1}^{n} x_i^2 x_j^2 \mathbb{1}_{\{x_i \geq 0\}} \mathbb{1}_{\{x_j \geq 0\}}\right].$$

Applying the same analysis to $\lambda_2(0)$, we have with probability $1 - 4\exp(-m/c')$,

$$\lambda_2(0)\|\boldsymbol{p}_2\|^2 = \frac{1}{m}\sum_{i=k}^{m}(u_{0,k}^2 + v_{0,k}^2)\left(\mathbb{1}_{\{u_k(0)\leq 0\}}\right)^2 \sum_{i,j=1}^{n} x_i^2 x_j^2 \mathbb{1}_{\{x_i\leq 0\}}\mathbb{1}_{\{x_j\leq 0\}}$$

$$\in \left[\frac{1}{2}\sum_{i,j=1}^{n} x_i^2 x_j^2 \mathbb{1}_{\{x_i\leq 0\}}\mathbb{1}_{\{x_j\leq 0\}}, \frac{3}{2}\sum_{i,j=1}^{n} x_i^2 x_j^2 \mathbb{1}_{\{x_i\leq 0\}}\mathbb{1}_{\{x_j\leq 0\}}\right].$$

The largest eigenvalue is $\max\{\lambda_1(0), \lambda_2(0)\}$. Combining the results together, we have with probability at least $1 - 4\exp(-m/c')$,

$$\frac{1}{2}M \leq \|K(0)\| \leq \frac{3}{2}M,$$

where $M = \max\left\{\dfrac{\sum_{i,j=1}^{n} x_i^2 x_j^2 \mathbb{1}\{x_i\geq 0\}\mathbb{1}\{x_j\geq 0\}}{\sum_{i=1}^{n} x_i^2 \mathbb{1}\{x_i\geq 0\}}, \dfrac{\sum_{i,j=1}^{n} x_i^2 x_j^2 \mathbb{1}\{x_i\leq 0\}\mathbb{1}\{x_j\leq 0\}}{\sum_{i=1}^{n} x_i^2 \mathbb{1}\{x_i\leq 0\}}\right\}.$ $\square$

## F  SCALE ANALYSIS FOR $R_\lambda$ AND $R_g$

**Proposition 3.** *Let $\rho := |g(t)|/\sqrt{m}$. Assume $m \gg 1$ and $|\lambda(t)| \leq C$ for some constant $C > 0$. If $\rho \ll 1$, i.e., $|g(t)| = o(\sqrt{m})$, then*

*(i)* $|R_g(t)| \leq \frac{\|\boldsymbol{x}\|^2\eta^2}{d}\rho^2 + \frac{\|\boldsymbol{x}\|^2\eta^2|y|}{d\sqrt{m}}\rho = o(1),$

*(ii)* $|R_\lambda(t)| \leq \frac{(\eta^2\lambda(t)-4\eta)\|\boldsymbol{x}\|^2}{d}\rho^2 + \left(\frac{2\eta^2\|\boldsymbol{x}\|^2|y|\lambda(t)+4\eta\|\boldsymbol{x}\|^2|y|}{d\sqrt{m}}\right)\rho + \frac{\eta^2\|\boldsymbol{x}\|^2 y^2\lambda(t)}{dm} = o(1).$

*Proof of Proposition 3.* According to dynamics equations, i.e., Eq. (11) and (12),

$$R_g(t) = \frac{\|\boldsymbol{x}\|^2}{md}\eta^2(g(t)-y)g(t),$$

$$R_\lambda(t) = \eta\frac{\|\boldsymbol{x}\|^2}{md}(g(t)-y)^2\left(\eta\lambda(t) - 4\frac{g(t)}{g(t)-y}\right).$$

Let $\rho = o(1)$. Then with simple application of the triangle inequality, we have

$$|R_g(t)| = \left|\frac{\|\boldsymbol{x}\|^2\eta^2 g(t)^2}{md} - \frac{\|\boldsymbol{x}\|^2\eta^2 g(t)y}{md}\right| \leq \frac{\|\boldsymbol{x}\|^2\eta^2}{d}\rho^2 + \frac{\|\boldsymbol{x}\|^2\eta^2|y|}{d\sqrt{m}}\rho = o(1),$$

and

$$|R_\lambda(t)| = \left|\frac{\|\boldsymbol{x}\|^2\eta^2}{md}(g(t)-y)^2\lambda(t) - \frac{4\|\boldsymbol{x}\|^2\eta}{md}g(t)(g(t)-y)\right|$$

$$= \left|\frac{\eta^2\lambda(t)\|\boldsymbol{x}\|^2 - 4\eta\|\boldsymbol{x}\|^2}{md}g(t)^2 + \frac{4\|\boldsymbol{x}\|^2\eta y - 2\|\boldsymbol{x}\|^2\eta^2 y\lambda(t)}{md}g(t) + \frac{\|\boldsymbol{x}\|^2\eta^2 y^2\lambda(t)}{md}\right|$$

$$\leq \frac{(\eta^2\lambda(t)-4\eta)\|\boldsymbol{x}\|^2}{d}\rho^2 + \left(\frac{2\eta^2\|\boldsymbol{x}\|^2|y|\lambda(t)+4\eta\|\boldsymbol{x}\|^2|y|}{d\sqrt{m}}\right)\rho + \frac{\eta^2\|\boldsymbol{x}\|^2 y^2\lambda(t)}{dm} = o(1).$$

$\square$

**Proposition 4.** *Let $\rho := |g(t)|/\sqrt{m}$ and assume $m \gg 1$. If $\rho \in [C_1, C_2]$ for some constants $C_1, C_2 > 0$, i.e., $|g(t)| = \Theta(\sqrt{m})$ then*

*(i)* $R_g(t) \in \left[\frac{\|\boldsymbol{x}\|^2\eta^2 C_1^2}{d} - \epsilon, \frac{\|\boldsymbol{x}\|^2\eta^2 C_2^2}{d} + \epsilon\right],$

*(ii)* *if $\lambda \leq 4/\eta$, $R_\lambda(t) \in \left[\frac{\|\boldsymbol{x}\|^2 C_2^2\eta(\eta\lambda(t)-4)}{d} - \epsilon, \frac{\|\boldsymbol{x}\|^2 C_1^2\eta(\eta\lambda(t)-4)}{d} + \epsilon\right],$*

 *otherwise, $R_\lambda(t) \in \left[\frac{\|\boldsymbol{x}\|^2 C_1^2\eta(\eta\lambda(t)-4)}{d} - \epsilon, \frac{\|\boldsymbol{x}\|^2 C_2 2\eta(\eta\lambda(t)-4)}{d} + \epsilon\right],$*

*where $\epsilon = O(1/\sqrt{m})$.*

*Proof of Proposition 4.* According to dynamics equations, i.e., Eq. (11) and (12),

$$R_g(t) = \frac{\|\boldsymbol{x}\|^2}{md}\eta^2(g(t)-y)g(t),$$

$$R_\lambda(t) = \eta\frac{\|\boldsymbol{x}\|^2}{md}(g(t)-y)^2\left(\eta\lambda(t) - 4\frac{g(t)}{g(t)-y}\right).$$

Let $C_1 \le \rho \le C_2$. Then with simple application of the triangle inequity, we have

$$R_g(t) = \frac{\|\boldsymbol{x}\|^2\eta^2 g(t)^2}{md} - \frac{\|\boldsymbol{x}\|^2\eta^2 g(t)y}{md} = \frac{\|\boldsymbol{x}\|^2\eta^2}{d}\rho^2 - \frac{\|\boldsymbol{x}\|^2\eta^2 y}{\sqrt{m}d}\rho.$$

Then

$$\frac{\|\boldsymbol{x}\|^2\eta^2}{d}C_1^2 - \frac{\|\boldsymbol{x}\|^2\eta^2 y}{\sqrt{m}d}C_2 \le R_g(t) \le \frac{\|\boldsymbol{x}\|^2\eta^2}{d}C_2^2 - \frac{\|\boldsymbol{x}\|^2\eta^2 y}{\sqrt{m}d}C_1.$$

And

$$R_\lambda(t) = \frac{\|\boldsymbol{x}\|^2\eta^2}{md}(g(t)-y)^2\lambda(t) - \frac{4\|\boldsymbol{x}\|^2\eta}{md}g(t)(g(t)-y)$$
$$= \frac{\eta^2\lambda(t)\|\boldsymbol{x}\|^2 - 4\eta\|\boldsymbol{x}\|^2}{d}\rho^2 + \frac{4\|\boldsymbol{x}\|^2\eta y - 2\|\boldsymbol{x}\|^2\eta^2 y\lambda(t)}{\sqrt{m}d}\rho + \frac{\|\boldsymbol{x}\|^2\eta^2 y^2\lambda(t)}{md}.$$

If $\lambda(t) \le 4/\eta$, we have

$$R_\lambda(t) \ge \frac{\eta^2\lambda(t)\|\boldsymbol{x}\|^2 - 4\eta\|\boldsymbol{x}\|^2}{d}C_2^2 - \frac{4\|\boldsymbol{x}\|^2\eta y + 2\|\boldsymbol{x}\|^2\eta^2 y\lambda(t)}{\sqrt{m}d}C_2,$$

$$R_\lambda(t) \le \frac{\eta^2\lambda(t)\|\boldsymbol{x}\|^2 - 4\eta\|\boldsymbol{x}\|^2}{d}C_1^2 + \frac{4\|\boldsymbol{x}\|^2\eta y + 2\|\boldsymbol{x}\|^2\eta^2 y\lambda(t)}{\sqrt{m}d}C_1 + \frac{\|\boldsymbol{x}\|^2\eta^2 y^2\lambda(t)}{md}.$$

If $\lambda(t) \ge 4/\eta$, we have

$$R_\lambda(t) \ge \frac{\eta^2\lambda(t)\|\boldsymbol{x}\|^2 - 4\eta\|\boldsymbol{x}\|^2}{d}C_1^2 - \frac{4\|\boldsymbol{x}\|^2\eta y + 2\|\boldsymbol{x}\|^2\eta^2 y\lambda(t)}{\sqrt{m}d}C_2,$$

$$R_\lambda(t) \le \frac{\eta^2\lambda(t)\|\boldsymbol{x}\|^2 - 4\eta\|\boldsymbol{x}\|^2}{d}C_2^2 + \frac{4\|\boldsymbol{x}\|^2\eta y + 2\|\boldsymbol{x}\|^2\eta^2 y\lambda(t)}{\sqrt{m}d}C_1 + \frac{\|\boldsymbol{x}\|^2\eta^2 y^2\lambda(t)}{md}.$$

Picking $\epsilon = \max\left(\frac{4\|\boldsymbol{x}\|^2\eta y + 2\|\boldsymbol{x}\|^2\eta^2 y\lambda(t)}{\sqrt{m}d}C_2 + \frac{\|\boldsymbol{x}\|^2\eta^2 y^2\lambda(t)}{md}, \frac{\|\boldsymbol{x}\|^2\eta^2 y}{\sqrt{m}d}C_2\right)$, we have the result.

$\square$

## G   SCALE ANALYSIS FOR $R_K$ AND $R_{\mathbf{g}}$

**Proposition 5.** *Let $\rho := \|\mathbf{g}(t)\|/\sqrt{m}$. Assume $m \gg 1$ and $\lambda_1(t) \le C$ for some constant $C > 0$. If $\rho \ll 1$, i.e. $\|\mathbf{g}(t)\| = o(\sqrt{m})$, then*

 (i) $\|R_{\mathbf{g}}(t)\| \le 2\eta^2(\rho^2 + \|\mathbf{y}\|\rho/\sqrt{m})\sum_i x_i^2 = o(1)$,

 (ii) $\|R_K(t)\| \le 2\eta^2 C(\rho^2 + \|\mathbf{y}\|\rho/\sqrt{m})\sum_i x_i^2 + 8\eta(\rho^2 + \|\mathbf{y}\|\rho/\sqrt{m})\sum_i x_i^2 = o(1)$.

*Proof of Proposition 5.* According to dynamics equations for multiple training examples, i.e., Eq. (14) and (15), we have

$$R_{\mathbf{g}}(t) = \frac{\eta^2}{m}(\mathbf{g}(t) - \mathbf{y}) \odot \boldsymbol{m}_+)^T (\mathbf{g}(t) \odot \boldsymbol{m}_+) \boldsymbol{p}_1 \boldsymbol{p}_1^T + \frac{\eta^2}{m}(\mathbf{g}(t) - \mathbf{y}) \odot \boldsymbol{m}_-)^T (\mathbf{g}(t) \odot \boldsymbol{m}_-) \boldsymbol{p}_2 \boldsymbol{p}_2^T,$$

$$R_K(t) = \frac{\eta^2}{m} ((\mathbf{g}(t) - \mathbf{y}) \odot \boldsymbol{m}_+)^T K(t) ((\mathbf{g}(t) - \mathbf{y}) \odot \boldsymbol{m}_+) \boldsymbol{p}_1 \boldsymbol{p}_1^T$$

$$+ \frac{\eta^2}{m} ((\mathbf{g}(t) - \mathbf{y}) \odot \boldsymbol{m}_-)^T K(t) ((\mathbf{g}(t) - \mathbf{y}) \odot \boldsymbol{m}_-) \boldsymbol{p}_2 \boldsymbol{p}_2^T$$

$$- \frac{4\eta}{m} ((\mathbf{g}(t) - \mathbf{y}) \odot \boldsymbol{m}_+)^T (\mathbf{g}(t) \odot \boldsymbol{m}_+) \boldsymbol{p}_1 \boldsymbol{p}_1^T$$

$$- \frac{4\eta}{m} ((\mathbf{g}(t) - \mathbf{y}) \odot \boldsymbol{m}_-)^T (\mathbf{g}(t) \odot \boldsymbol{m}_-) \boldsymbol{p}_2 \boldsymbol{p}_2^T.$$

Let $\rho = o(1)$. With simple application of the triangle inequality, we have

$$\|R_{\mathbf{g}}(t)\| \leq 2\eta^2(\rho^2 + \|\mathbf{y}\|\rho/\sqrt{m}) \sum_i x_i^2 = o(1),$$

since $\|\boldsymbol{p}_1\|^2 \leq \sum x_i^2$ and $\|\boldsymbol{p}_2\|^2 \leq \sum x_i^2$.

And

$$\|R_K(t)\| \leq 2\eta^2 C(\rho^2 + \|\mathbf{y}\|\rho/\sqrt{m}) \sum_i x_i^2 + 8\eta(\rho^2 + \|\mathbf{y}\|\rho/\sqrt{m}) \sum_i x_i^2 = o(1).$$

$\square$

**Proposition 6.** *Let* $\rho := \|\mathbf{g}(t)\|/\sqrt{m}$. *Assume* $m \gg 1$ *and* $\eta < 4/\lambda_1(t)$. *If* $\rho \in [C_1, C_2]$ *for some constants* $C_1, C_2 > 0$, *i.e.,* $\|\mathbf{g}(t)\| = \Theta(\sqrt{m})$, *then*

*(i)* $\|R_{\mathbf{g}}(t)\| \in \left[\min\{\|\boldsymbol{p}_1\|^2 C_1^2, \|\boldsymbol{p}_2\|^2 C_2^2\}\eta^2 + \epsilon, \max\{\|\boldsymbol{p}_1\|^2 C_1^2, \|\boldsymbol{p}_2\|^2 C_2^2\}\eta^2 + \epsilon\right],$

*(ii)* $\|R_K(t)\| \in \left[-\min\{\|\boldsymbol{p}_1\|^2 C_1^2, \|\boldsymbol{p}_2\|^2 C_2^2\}\eta\|K(t) - 4\eta I\| - \epsilon, -\max\{\|\boldsymbol{p}_1\|^2 C_1^2, \|\boldsymbol{p}_2\|^2 C_2^2\}\eta\|K(t) - 4\eta I\| + \epsilon\right],$

*where* $\epsilon = O(1/\sqrt{m})$.

*Proof of Proposition 6.* According to dynamics equations for multiple training examples, i.e., Eq. (14) and (15), we have

$$R_{\mathbf{g}}(t) = \frac{\eta^2}{m}(\mathbf{g}(t) - \mathbf{y}) \odot \boldsymbol{m}_+)^T (\mathbf{g}(t) \odot \boldsymbol{m}_+) \boldsymbol{p}_1 \boldsymbol{p}_1^T + \frac{\eta^2}{m}(\mathbf{g}(t) - \mathbf{y}) \odot \boldsymbol{m}_-)^T (\mathbf{g}(t) \odot \boldsymbol{m}_-) \boldsymbol{p}_2 \boldsymbol{p}_2^T,$$

$$R_K(t) = \frac{\eta^2}{m} ((\mathbf{g}(t) - \mathbf{y}) \odot \boldsymbol{m}_+)^T K(t) ((\mathbf{g}(t) - \mathbf{y}) \odot \boldsymbol{m}_+) \boldsymbol{p}_1 \boldsymbol{p}_1^T$$

$$+ \frac{\eta^2}{m} ((\mathbf{g}(t) - \mathbf{y}) \odot \boldsymbol{m}_-)^T K(t) ((\mathbf{g}(t) - \mathbf{y}) \odot \boldsymbol{m}_-) \boldsymbol{p}_2 \boldsymbol{p}_2^T$$

$$- \frac{4\eta}{m} ((\mathbf{g}(t) - \mathbf{y}) \odot \boldsymbol{m}_+)^T (\mathbf{g}(t) \odot \boldsymbol{m}_+) \boldsymbol{p}_1 \boldsymbol{p}_1^T$$

$$- \frac{4\eta}{m} ((\mathbf{g}(t) - \mathbf{y}) \odot \boldsymbol{m}_-)^T (\mathbf{g}(t) \odot \boldsymbol{m}_-) \boldsymbol{p}_2 \boldsymbol{p}_2^T.$$

Note that $\mathbf{g}(t) \odot \boldsymbol{m}_+ + \mathbf{g}(t) \odot \boldsymbol{m}_- = \mathbf{g}(t)$. We further denote $\rho_+ := \|\mathbf{g}(t) \odot \boldsymbol{m}_+\|/\sqrt{m}$ and $\rho_- := \|\mathbf{g}(t) \odot \boldsymbol{m}_-\|/\sqrt{m}$. Then it is not hard to see that $\rho_+^2 + \rho_-^2 = \rho^2$. And we have

$$R_{\mathbf{g}}(t) = \eta^2/m\|\mathbf{g}(t) \odot \boldsymbol{m}_+\|^2 \boldsymbol{p}_1 \boldsymbol{p}_1^T + \eta^2/m\|\mathbf{g}(t) \odot \boldsymbol{m}_-\|^2 \boldsymbol{p}_2 \boldsymbol{p}_2^T - \eta^2/m(\mathbf{y} \odot \boldsymbol{m}+)^T (\mathbf{g}(t) \odot \boldsymbol{m}_+) \boldsymbol{p}_1 \boldsymbol{p}_1^T$$

$$- \eta^2/m(\mathbf{y} \odot \boldsymbol{m}-)^T (\mathbf{g}(t) \odot \boldsymbol{m}_-) \boldsymbol{p}_2 \boldsymbol{p}_2^T.$$

Therefore

$$\min\{\|\boldsymbol{p}_1\|^2 C_1^2, \|\boldsymbol{p}_2\|^2 C_2^2\}\eta^2 + O\left(\frac{1}{\sqrt{m}}\right) \leq \|R_{\mathbf{g}}(t)\| \leq \max\{\|\boldsymbol{p}_1\|^2 C_1^2, \|\boldsymbol{p}_2\|^2 C_2^2\}\eta^2 + O\left(\frac{1}{\sqrt{m}}\right).$$

For $R_K(t)$, since the top eigenvalue of $K(t) - 4\eta I$ is negative by our assumption, we have

$$\|R_K(t)\| \leq -\min\{\|\boldsymbol{p}_1\|^2 C_1^2, \|\boldsymbol{p}_2\|^2 C_2^2\}\eta\|K(t) - 4\eta I\| + O(1/\sqrt{m}),$$
$$\|R_K(t)\| \geq -\max\{\|\boldsymbol{p}_1\|^2 C_1^2, \|\boldsymbol{p}_2\|^2 C_2^2\}\eta\|K(t) - 4\eta I\| - O(1/\sqrt{m}).$$

$\square$

# H PROOF OF THEOREM 2 AND ANALYSIS ON OPTIMIZATION DYNAMICS FOR MULTIPLE TRAINING EXAMPLES

By Eq. (5), the tangent kernel $K$ at step $t$ is defined as:

$$K_{i,j}(t) = \langle \nabla_{\mathbf{v}} g_i(t), \nabla_{\mathbf{v}} g_j(t) \rangle + \langle \nabla_{\mathbf{u}} g_i(t), \nabla_{\mathbf{u}} g_j(t) \rangle$$
$$= \frac{1}{m} \sum_{k=1}^{m} \left( (u_k(t))^2 + (v_k(t))^2 \right) x_i x_j \mathbb{1}_{\{u_k(0)x_i \geq 0\}} \mathbb{1}_{\{u_k(0)x_j \geq 0\}}, \quad \forall i, j \in [n].$$

Similar to single example case, the largest eigenvalue of of tangent kernel is bounded from 0:

**Proposition 7.** *For any $\delta \in (0,1)$, if $m \geq c' \log(4/\delta)$ where $c'$ is an absolute constant, with probability at least $1 - \delta$, $M/2 \leq \lambda_{\max}(K(0)) \leq 3M/2$ where $M = \max\left\{ \frac{\sum_{i,j=1}^{n} x_i^2 x_j^2 \mathbb{1}_{\{x_i \geq 0\}} \mathbb{1}_{\{x_j \geq 0\}}}{\sum_{i=1}^{n} x_i^2 \mathbb{1}_{\{x_i \geq 0\}}}, \frac{\sum_{i,j=1}^{n} x_i^2 x_j^2 \mathbb{1}_{\{x_i \leq 0\}} \mathbb{1}_{\{x_j \leq 0\}}}{\sum_{i=1}^{n} x_i^2 \mathbb{1}_{\{x_i \leq 0\}}} \right\}.$*

The proof can be found in Appendix E.

For the simplicity of notation, given $\boldsymbol{p}, \boldsymbol{m} \in \mathbb{R}^n$, we define the matrices $K_{\boldsymbol{p},\boldsymbol{m}}$ and $Q_{\boldsymbol{p},\boldsymbol{m}}$:

$$K_{\boldsymbol{p},\boldsymbol{m}}(t) := \left( (\mathbf{g}(t) - \mathbf{y}) \odot \boldsymbol{m} \right)^T K(t) \left( (\mathbf{g}(t) - \mathbf{y}) \odot \boldsymbol{m} \right) \boldsymbol{p}\boldsymbol{p}^T,$$
$$Q_{\boldsymbol{p},\boldsymbol{m}}(t) := \left( (\mathbf{g}(t) - \mathbf{y}) \odot \boldsymbol{m} \right)^T \left( \mathbf{g}(t) \odot \boldsymbol{m} \right) \boldsymbol{p}\boldsymbol{p}^T$$

It is not hard to see that for all $t$, $K_{\boldsymbol{p},\boldsymbol{m}}$ and $Q_{\boldsymbol{p},\boldsymbol{m}}$ are rank-1 matrices. Specially, $\boldsymbol{p}$ is the only eigenvector of $K_{\boldsymbol{p},\boldsymbol{m}}$ and $Q_{\boldsymbol{p},\boldsymbol{m}}$.

With the above notations, we can write the update equations for $\mathbf{g}(t) - \mathbf{y}$ and $K(t)$ during gradient descent with learning rate $\eta$:

**Dynamics equations.**

$$\mathbf{g}(t+1) - \mathbf{y} = \left( I - \eta K(t) + \underbrace{\frac{\eta^2}{m} \left( Q_{\boldsymbol{p}_1,\boldsymbol{m}_+}(t) + Q_{\boldsymbol{p}_2,\boldsymbol{m}_-}(t) \right)}_{R_{\mathbf{g}}(t)} \right) (\mathbf{g}(t) - \mathbf{y}), \quad (14)$$

$$K(t+1) = K(t) + \underbrace{\frac{\eta^2}{m} \left( K_{\boldsymbol{p}_1,\boldsymbol{m}_+}(t) + K_{\boldsymbol{p}_2,\boldsymbol{m}_-}(t) \right) - \frac{4\eta}{m} \left( Q_{\boldsymbol{p}_1,\boldsymbol{m}_+}(t) + Q_{\boldsymbol{p}_2,\boldsymbol{m}_-}(t) \right)}_{R_K(t)}, \quad (15)$$

where $\boldsymbol{m}_+, \boldsymbol{m}_- \in \mathbb{R}^n$ are mask vectors:

$$m_{+,i} = \mathbb{1}_{\{i \in \mathcal{S}_+\}}, \quad m_{-,i} = \mathbb{1}_{\{i \in \mathcal{S}_-\}}.$$

Now we are ready to discuss different three optimization dynamics for multiple training examples case, similar to single training example case in the following.

**Monotonic convergence: sub-critical learning rates ($\eta < 2/\lambda_1(0)$).** We use the key observation that when $\|\mathbf{g}(t)\|$ is small, i.e., $o(\sqrt{m})$, and $\|K(t)\|$ is bounded, then $\|R_{\mathbf{g}}(t)\|$ and $\|R_K(t)\|$ are of

the order $o(1)$ (see a formal statement Proposition 5 in Appendix G). Then the dynamics equations approximately reduce to the ones of linear dynamics for multiple training examples

$$\mathbf{g}(t+1) - \mathbf{y} = (I - \eta K(t) + o(1)) (\mathbf{g}(t) - \mathbf{y}),$$
$$K(t+1) = K(t) + o(1).$$

At initialization, $\|\mathbf{g}(0)\| = O(1)$ with high probability over random initialization. By Proposition 5, when $\|\mathbf{g}(t)\| = o(\sqrt{m})$, the optimization follows linear dynamics. By the choice of the learning rate, we will have for all $t \geq 0$, $\|I - \eta K(t)\| < 2$, hence $\|\mathbf{g}(t) - \mathbf{y}\|$ decreases exponentially. The cumulative change on the norm of tangent kernel is $o(1)$ since $\|R_K(t)\| = o(1)$ and the loss decreases exponentially hence $\sum \|R_K(t)\| = o(1) \cdot \log O(1) = o(1)$.

**Catapult convergence: super-critical learning rates** $(2/\lambda_1(0) < \eta < \min\{2/\lambda_2(0), 4/\lambda_1(0)\})$.

**Restate Theorem 2:** *Consider training a NQM Eq. (10) with squared loss on multiple training examples by gradient descent. Under Assumption 1, if the learning rate is super-critical i.e., $2/\lambda_1(0) < \eta < \min\{2/\lambda_2(0), 4/\lambda_1(0)\}$, then there exist $T_1, T_2, T_3, T_4$ such that $0 < T_1 < T_2 < T_3 < T_4$ and the training dynamics exhibits:*

(i) **Increasing phase:** $t \in [0, T_1]$. *In this phase, $\mathcal{L}(t) = o(m)$. The loss grows exponentially; both eigenvalues are nearly constant i.e., $|\lambda_k(t) - \lambda_k(0)| = o(1)$ for $k = 1, 2$.*

(ii) **Peak phase:** $t \in [T_2, T_3]$. *In this phase, $\mathcal{L}(t) = \Theta(m)$. For the tangent kernel:*

    (a) *If $2/\lambda_2(0) \leq 4/\lambda_1(0)$, both eigenvalues decrease significantly, i.e., $\lambda_1(t+1) - \lambda_1(t) < 0$, $\lambda_2(t+1) - \lambda_2(t) < 0$ and both difference are of the order $\Theta(1)$.*

    (b) *If $2/\lambda_2(0) > 4/\lambda_1(0)$, only $\lambda_1(t)$ decreases significantly.*

(iii) **Decreasing phase:** $t \in [T_4, \infty)$. *In this phase, the loss satisfies $\mathcal{L}(t) = o(m)$ again and decreases. Both eigenvalues are nearly constant until convergence: $|\lambda_k(t) - \lambda_k(\infty)| = o(1)$, for $k = 1, 2$.*

*Furthermore, $T_1 = o(\log m)$, $T_2 = \Theta(\log m)$, $T_3 - T_2 = \Theta(1)$ and $T_4 = \Theta(\log m)$.*

*Proof of Theorem 2.* We assume $2/\lambda_2(0) \leq 4/\lambda_1(0)$, since in this scenario, the catapult phase happens in both directions $\boldsymbol{p}_1, \boldsymbol{p}_2$, i.e., $\mathbf{g}(t)$ has significant projection in both directions at the peak of the loss. In fact, as $\boldsymbol{p}_1$ is orthogonal to $\boldsymbol{p}_2$, the training dynamics in two directions are almost independent to each other due to the special structure of the Hessian. If instead $2/\lambda_2(0) > 4/\lambda_1(0)$, the catapult phase mainly happens in the direction $\boldsymbol{p}_1$ since in the direction of $\boldsymbol{p}_2$ the linear dynamics dominate. In this simpler setting, the analysis will be implied by our following analysis.

*(i) Increasing phase.* Initially $\|\mathbf{g}(t) - \mathbf{y}\|$ grows exponentially following linear dynamics by Proposition 5. By the choice of the learning rate, we will have for $t$ in the increasing phase,

$$\left| \boldsymbol{p}_1^T (I - \eta K(t)) \boldsymbol{p}_1 \right| / \|\boldsymbol{p}_1\|^2 = |2 - \eta \lambda_1(t)| = |2 - \eta \lambda_1(0) + o(1)| > 1.$$

And following the same analysis, we have $|\boldsymbol{p}_2^T (I - \eta K(t)) \boldsymbol{p}_2| / \|\boldsymbol{p}_2\|^2 > 1$ as well. Therefore, $\|\mathbf{g}(t) - \mathbf{y}\|$ increases in the direction $\boldsymbol{p}_1$ and $\boldsymbol{p}_2$ (If instead $2/\lambda_2(0) > 4/\lambda_1(0)$, $|\boldsymbol{p}_2^T (I - \eta K(t)) \boldsymbol{p}_2| / \|\boldsymbol{p}_2\|^2 < 1$ therefore $\|\mathbf{g}(t) - \mathbf{y}\|$ only grows in the direction $\boldsymbol{p}_1$). Note that when both $|\boldsymbol{p}_1^T (I - \eta K(t)) \boldsymbol{p}_1| / \|\boldsymbol{p}_1\|^2$ and $|\boldsymbol{p}_2^T (I - \eta K(t)) \boldsymbol{p}_2| / \|\boldsymbol{p}_2\|^2$ are smaller than 1, the loss stops increasing hence enters the peak phase.

Up to the iteration when $\|\mathbf{g}(t)\| = o(\sqrt{m})$, the cumulative change of $\lambda_1(t)$ and $\lambda_2(t)$ from initialization will be $o(1)$ since $\|\mathbf{g}(t)\|$ increases exponentially. Specifically, for $\lambda_1(t)$

$$|\lambda_1(t) - \lambda_1(0)| = \sum_{t=0}^{T_1} |\boldsymbol{p}_1^T R_K(t) \boldsymbol{p}_1| / \|\boldsymbol{p}_1\|^2 = \sum_{t=0}^{T_2} \Theta \left( \|\mathbf{g}(t)\|^2 / m \right) = o(1).$$

The same analysis can be applied to $\lambda_2(t)$ as well which gives $|\lambda_2(t) - \lambda_2(0)| = o(1)$.

Furthermore, it is not hard to see that until the loss grows to the order of $\Theta(m)$, the tangent kernel does not change much hence the loss keeps increasing exponentially.

*(ii) Peak phase.* We will use the key observation that when $\|\mathbf{g}(t)\|$ is large, i.e., of the order $\Theta(\sqrt{m})$, $\|R_K(t)\|$ and $\|R_\mathbf{g}(t)\|$ will be of the order $\Theta(1)$ (See the formal statement Proposition 6 in Appendix G), which can lead to the decrease of the loss.

When $\|\mathbf{g}(t)\|$ increases to the order of $\Theta(\sqrt{m})$, since $\|\mathbf{g}(t)\|$ only grows in direction $\boldsymbol{p}_1$ and $\boldsymbol{p}_2$, we can see that $\mathbf{g}(t)$ is mostly aligned with $\boldsymbol{p}_1$ and $\boldsymbol{p}_2$, i.e., $\mathbf{g}(t)^T\boldsymbol{p}_1/\|\boldsymbol{p}_1\| + \mathbf{g}(t)^T\boldsymbol{p}_2/\|\boldsymbol{p}_2\| \approx \|\mathbf{g}(t)\|$. Here by our assumption $\|\mathbf{y}\| = O(1)$ which is small compared to the scale of $\|\mathbf{g}(t)\|$ hence we can omit it. Since $\eta$ is initially chosen to be in the interval $(2/\lambda_1(0), 4/\lambda_1(0))$ and $\lambda_1(t) = \lambda_1(0) + o(1)$ when $\|\mathbf{g}(t)\| = o(\sqrt{m})$, we in general have $\lambda_1(t) < 4/\eta$ at the begining of the peak phase. Therefore, by Proposition 6, the tangent kernel decreases significantly in direction $\boldsymbol{p}_1$ i.e., $\lambda_1(t)$ decreases. Specifically, we have

$$\boldsymbol{p}_1^T K(t+1)\boldsymbol{p}_1 = \boldsymbol{p}_1^T K(t)\boldsymbol{p}_1 + \boldsymbol{p}_1^T R_K(t)\boldsymbol{p}_1 < \boldsymbol{p}_1^T K(t)\boldsymbol{p}_1,$$

where $|\boldsymbol{p}_1^T R_K(t)\boldsymbol{p}_1| = \Theta(1)$ and $\boldsymbol{p}_1^T R_K(t)\boldsymbol{p}_1 < 0$. The same analysis works for the direction $\boldsymbol{p}_2$ then we have $\boldsymbol{p}_2^T K(t+1)\boldsymbol{p}_2 < \boldsymbol{p}_2^T K(t)\boldsymbol{p}_2$, i.e., $\lambda_2(t+1) < \lambda_2(t)$.

Similarly, as $\|\mathbf{g}(t)\|$ increases, $\|R_\mathbf{g}(t)\|$ increases as well by Proposition 6. Therefore, the factor $I - \eta K(t) + R_\mathbf{g}(t)$ on $\mathbf{g}(t) - \mathbf{y}$ decreases which slows down the increase of the loss.

In general, the loss grows exponentially prior to the peak phase hence $T_2 = \Theta(\log m)$. And the peak phase only lasts $\Theta(1)$ steps during training, i.e., $|T_3 - T_2| = \Theta(1)$, since the decrease of $\lambda_1(t)$ and $\lambda_2(t)$ is $\Theta(1)$ at each step.

*(iii) Decreasing phase.* With the tangent kernel $K(t)$ decreasing and $\|\mathbf{g}(t) - \mathbf{y}\|$ increasing in the direction $\boldsymbol{p}_1$ and $\boldsymbol{p}_2$, the factor $I - \eta K(t) + R_\mathbf{g}(t)$ will ultimately smaller than 1 in both directions, which makes the peak phase ends since $\|\mathbf{g}(t) - \mathbf{y}\|$ starts to decrease. Again, due to the large scale of $\|\mathbf{g}(t)\|$, i.e. $\Theta(\sqrt{m})$, the tangent kernel still decreases significantly.

Similar to the single training example case, the decrease of $\lambda_1(t)$ and $\lambda_2(t)$ stops when $\|\mathbf{g}(t)\|$ decreases to the order $o(\sqrt{m})$, as $\|R_K(t)\| = o(1)$. In general we have $\|I - \eta K(t)\|$ smaller than 1 in both directions. Hence by Proposition 5, the training dynamics become linear that the loss decreases monotonically and the tangent kernel is nearly constant until convergence.

$\square$

**Divergence:** $(\eta > \eta_{\max} = 4/\lambda_1(0))$. Similar to the increasing phase in the catapult convergence, initially $\|\mathbf{g}(t) - \mathbf{y}\|$ increases in direction $\boldsymbol{p}_1$ and $\boldsymbol{p}_2$ since linear dynamics dominate and the learning rate is chosen to be larger than $\eta_{\mathrm{crit}}$. Also, we approximately have $\eta > 4/\lambda_1(t)$ at the end of the increasing phase, by a similar analysis for the catapult convergence. We consider the evolution of $K(t)$ in the direction $\boldsymbol{p}_1$. Note that when $\|\mathbf{g}(t)\|$ increases to the order of $\Theta(\sqrt{m})$, $\mathbf{g}(t) \odot \boldsymbol{m}_+$ will be aligned with $\boldsymbol{p}_1$, hence with simple calculation, we approximately have

$$\boldsymbol{p}_1^T R_K(t)\boldsymbol{p}_1 \approx \frac{\|\mathbf{g}(t)\|^2\|\boldsymbol{p}_1\|^2}{m}\eta(\lambda_1(t) - 4\eta) > 0.$$

Therefore, $\lambda_1(t)$ increases since $\boldsymbol{p}_1^T K(t+1)\boldsymbol{p}_1 = \boldsymbol{p}_1^T K(t)\boldsymbol{p}_1 + \boldsymbol{p}_1^T R_K(t)\boldsymbol{p}_1 > \boldsymbol{p}_1^T K(t)\boldsymbol{p}_1$. As a result, $\|I - \eta K(t) + R_\mathbf{g}(t)\|$ becomes even larger which makes $\|\mathbf{g}(t) - \mathbf{y}\|$ grows faster, and ultimately leads to divergence of the optimization.

# I   SPECIAL CASE OF QUADRATIC MODELS WHEN $\phi(\boldsymbol{x}) = 0$

In this section we will show under some special settings, the catapult phase phenomenon also happens and how two layer linear neural networks fit in our quadratic model.

We consider one training example $(\boldsymbol{x}, y)$ with label $y = 0$ and assume the initial tangent kernel $\lambda(0) = \Omega(1)$. Letting the feature vector $\phi(\boldsymbol{x}) = 0$, the quadratic model Eq.(3) becomes:

$$g(\mathbf{w}) = \frac{1}{2}\gamma\mathbf{w}^T \Sigma(\boldsymbol{x})\mathbf{w}.$$

For this quadratic model, we have the following proposition:

**Proposition 8.** *With learning rate* $\frac{2}{\lambda(0)} < \eta < \frac{4}{\lambda(0)}$, *if* $\Sigma(\boldsymbol{x})^2 = \|\boldsymbol{x}\|^2 \cdot I$, $g(\mathbf{w})$ *exhibits catapult phase.*

*Proof.* With simple computation, we get

$$g(t+1) = \left(1 - \eta\lambda(t) + \gamma\eta^2\|\boldsymbol{x}\|^2(g(t))^2\right)g(t),$$
$$\lambda(t+1) = \lambda(t) + \gamma\|\boldsymbol{x}\|^2(g(t))^2(\eta\lambda(t) - 4).$$

We note that the evolution of $g$ and $\lambda$ is almost the same with Eq. (11) and Eq. (12) if we regard $\gamma = 1/m$. Hence we can apply the same analysis to show the catapult phase phenomenon. $\square$

It is worth pointing out that the two-layer linear neural network with input $\boldsymbol{x} \in \mathbb{R}^d$ analyzed in Lewkowycz et al. (2020) that

$$f(\mathbf{U}, \mathbf{v}; x) = \frac{1}{\sqrt{m}}\mathbf{v}^T\mathbf{U}\boldsymbol{x},$$

where $\mathbf{v} \in \mathbb{R}^m, \mathbf{U} \in \mathbb{R}^{m \times d}$ is a special case of our model with $\mathbf{w} = \left[\text{Vec}(\mathbf{U})^T, \mathbf{v}^T\right]^T, \gamma = 1/\sqrt{m}$ and

$$\Sigma = \begin{pmatrix} 0 & I_m \otimes \boldsymbol{x} \\ I_m \otimes \boldsymbol{x}^T & 0 \end{pmatrix} \in \mathbb{R}^{md+m}.$$

## J  EXPERIMENTAL SETTINGS AND ADDITIONAL RESULTS

### J.1  VERIFICATION OF NON-LINEAR TRAINING DYNAMICS OF NQMS, I.E., FIGURE 3

We train the NQM which approximates the two-layer fully-connected neural network with ReLU activation function on 128 data points where each input is drawn i.i.d. from $\mathcal{N}(-2, 1)$ if the label is $-1$ or $\mathcal{N}(2, 1)$ if the label is 1. The network width is $5,000$.

### J.2  EXPERIMENTS FOR TRAINING DYNAMICS OF WIDE NEURAL NETWORKS WITH MULTIPLE EXAMPLES.

We train a two-layer fully-connected neural network with ReLU activation function on 128 data points where each input is drawn i.i.d. from $\mathcal{N}(-2, 1)$ if the label is $-1$ or $\mathcal{N}(2, 1)$ if the label is 1. The network width is $5,000$. See the results in Figure 5.

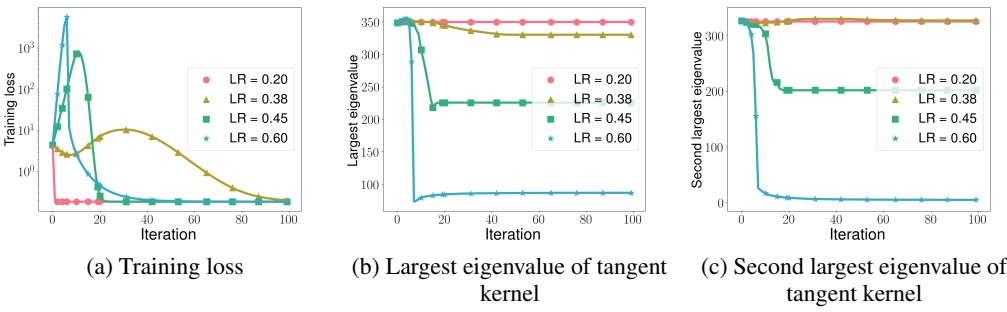

(a) Training loss        (b) Largest eigenvalue of tangent kernel        (c) Second largest eigenvalue of tangent kernel

Figure 5: **Training dynamics of wide neural networks for multiple examples case with different learning rates.** Compared to the training dynamics of NQMs, i.e., Figure 3, the behaviour of of top eigenvalues is almost the same with different learning rates: when $\eta < 0.37$, the kernel is nearly constant; when $0.37 < \eta < 0.39$, only $\lambda_1(t)$ decreases; when $0.39 < \eta < \eta_{\max}$, both $\lambda_1(t)$ and $\lambda_2(t)$ decreases. See the experiment setting in Appendix J.2.

### J.3 TRAINING DYNAMICS OF GENERAL QUADRATIC MODELS AND NEURAL NETWORKS.

As discussed at the end of Section 3, a more general quadratic model can exhibit the catapult phase phenomenon. Specifically, we consider a general quadratic model:

$$g(\mathbf{w}; \boldsymbol{x}) = \mathbf{w}^T \phi(\boldsymbol{x}) + \frac{1}{2}\gamma \mathbf{w}^T \Sigma(\boldsymbol{x})\mathbf{w}.$$

We will train the general quadratic model with different learning rates, and different $\gamma$ respectively, to see how the catapult phase phenomenon depends on these two factors. For comparison, we also implement the experiments for neural networks. See the experiment setting in the following:

**General quadratic models.** We set the dimension of the input $d = 100$. We let the feature vector $\phi(\boldsymbol{x}) = \boldsymbol{x}/\|\boldsymbol{x}\|$ where $x_i \sim \mathcal{N}(0, 1)$ i.i.d. for each $i \in [d]$. We let $\Sigma$ be a diagonal matrix with $\Sigma_{i,i} \in \{-1, 1\}$ randomly and independently. The weight parameters $\mathbf{w}$ are initialized by $\mathcal{N}(0, I_d)$. Unless stated otherwise, $\gamma = 10^{-3}$, and the learning rate is set to be 2.8.

**Neural networks.** We train a two-layer fully-connected neural networks with ReLU activation function on 20 data points of CIFAR-2. Unless stated otherwise, the network width is $10^4$, and the learning rate is set to be 2.8.

See the results in Figure 6.

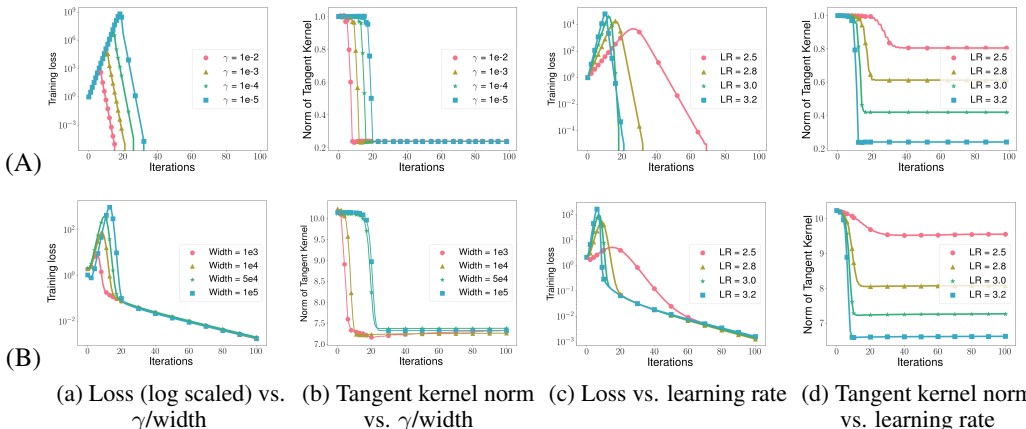

(a) Loss (log scaled) vs. (b) Tangent kernel norm (c) Loss vs. learning rate (d) Tangent kernel norm
    $\gamma$/width              vs. $\gamma$/width                                    vs. learning rate

Figure 6: **General quadratic models have similar training dynamics with neural networks when trained with super-critical learning rates.** Panel (A): experiments on general quadratic models. Smaller $\gamma$ or larger learning rates lead to larger training loss at the peak. Larger learning rates make tangent kernel decrease more. Panel (B): experiments on two-layer neural networks. Larger width (corresponding to smaller $\gamma$) and larger learning rates have similar effect on the training loss at the peak and decrease of tangent kernel norm with quadratic models. Note that width or $\gamma$ seems to have no effect on the tangent kernel norm at convergence.

### J.4 TEST PERFORMANCE OF $f$, $f_{\text{lin}}$ AND $f_{\text{quad}}$, I.E., FIGURE 2(B) AND FIGURE 4

For the architectures of two-layer fully connected neural network and two-layer convolutional neural network, we set the width to be $5,000$ and $1,000$ respectively. Specific to Figure 2(b), we use the architecture of a two-layer fully connected neural network.

Due to the large number of parameters in NQMs, we choose a small subset of all the datasets. We use the first class (airplanes) and third class (birds) of CIFAR-10, which we call CIFAR-2, and select 256 data points out of it as the training set. We use the number 0 and 2 of SVHN, and select 256 data points as the training set. We select $128, 256, 128$ data points out of MNIST, FSDD and AG NEWS dataset respectively as the training sets. The size of testing set is $2,000$ for all. When implementing SGD, we choose batch size to be 32.

For each setting, we report the average result of 5 independent runs.

### J.5 TEST PERFORMANCE OF $f$, $f_{\mathrm{lin}}$ AND $f_{\mathrm{quad}}$ IN TERMS OF ACCURACY

In this section, we report the best test accuracy for $f$, $f_{\mathrm{lin}}$ and $f_{\mathrm{quad}}$ corresponding to the best test loss in Figure 4. We use the same setting as in Appendix J.4.

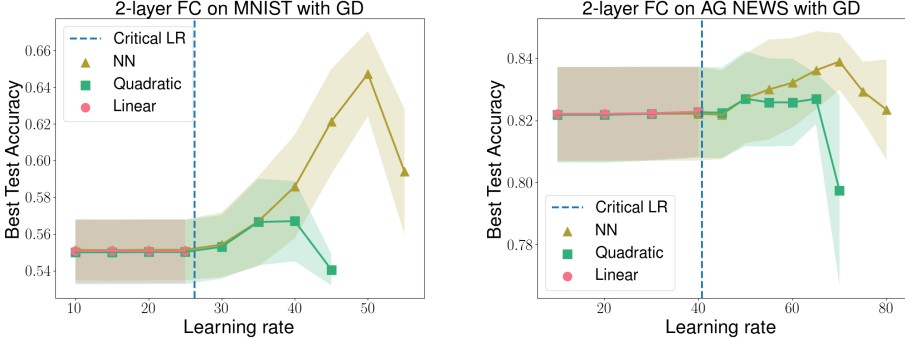

Figure 7: **Best test accuracy plotted against different learning rates for $f_{\mathrm{quad}}$, $f$, and $f_{\mathrm{lin}}$.** Left panel: 2-layer FC on MNIST trained with GD. Right panel: 2-layer FC on AG NEWS trained with GD.

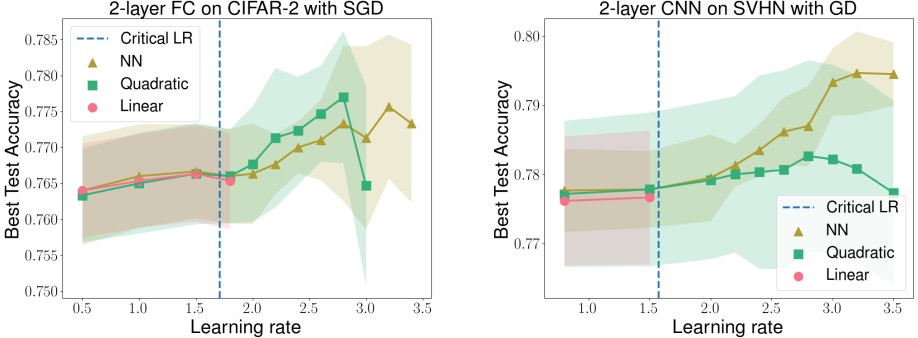

Figure 8: **Best test accuracy plotted against different learning rates for $f_{\mathrm{quad}}$, $f$, and $f_{\mathrm{lin}}$.** Left panel: 2-layer FC on CIFAR-2 trained with SGD. Right panel: 2-layer CNN on SVHN trained with GD.

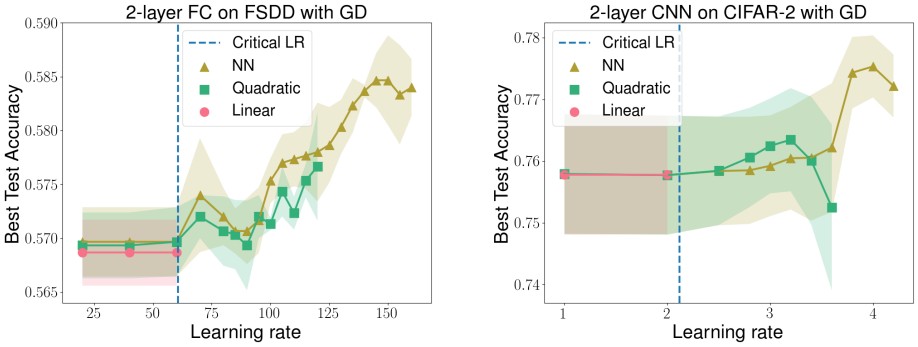

Figure 9: **Best test accuracy plotted against different learning rates for $f_{\mathrm{quad}}$, $f$, and $f_{\mathrm{lin}}$.** Left panel: 2-layer FC on FSDD trained with GD. Right panel: 2-layer CNN on CIFAR-2 trained with GD.

## J.6 TEST PERFORMANCE OF $f$, $f_{\text{lin}}$ AND $f_{\text{quad}}$ WITH ARCHITECTURE OF 3-LAYER FC

In this section, we extend our results for shallow neural networks discussed in Section 4 to 3-layer fully connected neural networks. In the same way, we compare the test performance of three models, $f$, $f_{\text{lin}}$ and $f_{\text{quad}}$ upon varying learning rate. We observe the same phenomenon for 3-layer ReLU activated FC with shallow neural networks. See Figure 10 and 11.

We use the first class (airplanes) and third class (birds) of CIFAR-10, which we call CIFAR-2, and select 100 data points out of it as the training set. We use the number 0 and 2 of SVHN, and select 100 data points as the training set. We select 100 data points out of AG NEWS dataset as the training set. For the speech data set FSDD, we select 100 data points in class 1 and 3 as the training set. The size of testing set is 500 for all.

For each setting, we report the average result of 5 independent runs.

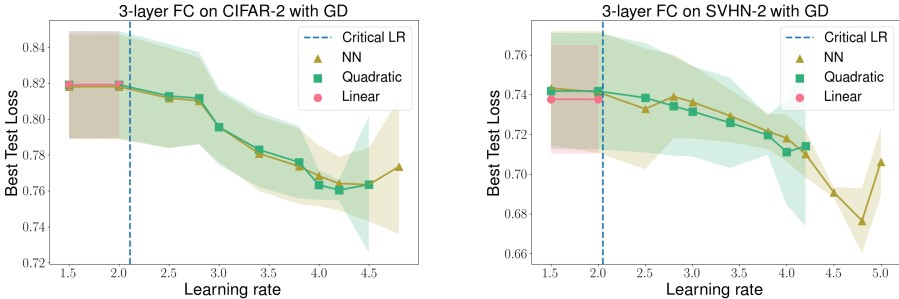

Figure 10: **Best test accuracy plotted against different learning rates for $f_{\text{quad}}$, $f$, and $f_{\text{lin}}$.** Left panel: 3-layer FC on CIFAR-2 trained with GD. Right panel: 3-layer FC on SVHN-2 trained with GD.

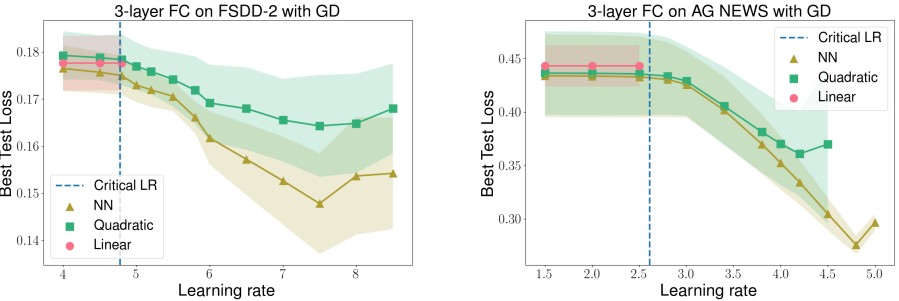

Figure 11: **Best test accuracy plotted against different learning rates for $f_{\text{quad}}$, $f$, and $f_{\text{lin}}$.** Left panel: 3-layer FC on FSDD-2 trained with GD. Right panel: 3-layer FC on AG NEWS trained with GD.

## J.7 TEST PERFORMANCE WITH TANH AND SWISH ACTIVATION FUNCTIONS

We replace ReLU by Tanh and Swish activation functions to train the models with the same setting as Figure 4. We observe the same phenomenon as we describe in Section 4.

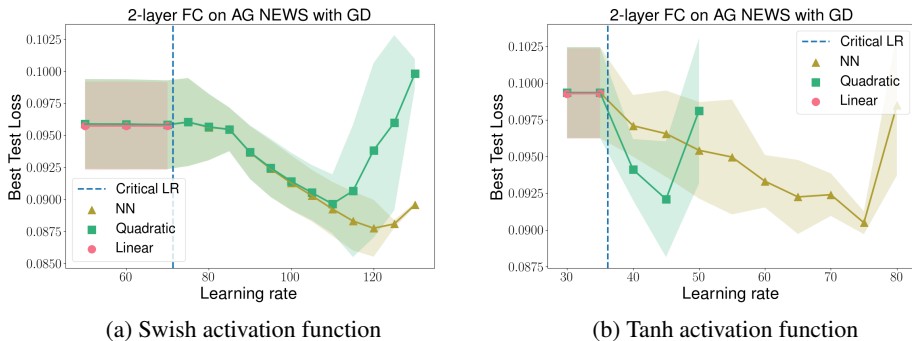

(a) Swish activation function        (b) Tanh activation function

Figure 12: **Best test loss plotted against different learning rates for** $f_{\text{quad}}$, $f$**, and** $f_{\text{lin}}$**.** We choose 2-layer FC as the architecture and train the models on AG NEWS with GD.

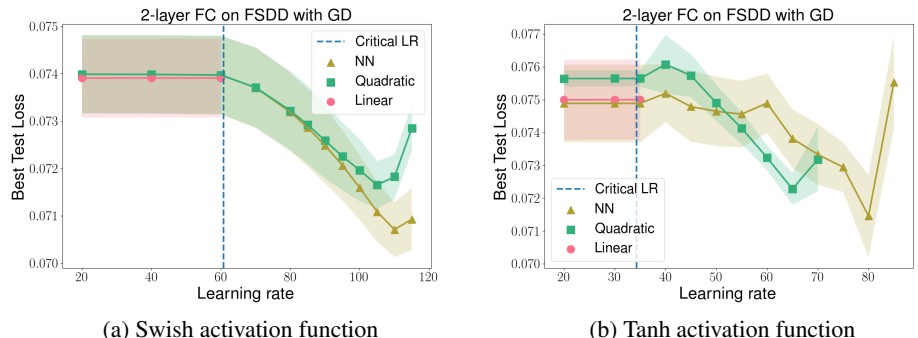

(a) Swish activation function        (b) Tanh activation function

Figure 13: **Best test loss plotted against different learning rates for** $f_{\text{quad}}$, $f$**, and** $f_{\text{lin}}$**.** We choose 2-layer FC as the architecture and train the models on FSDD with GD.

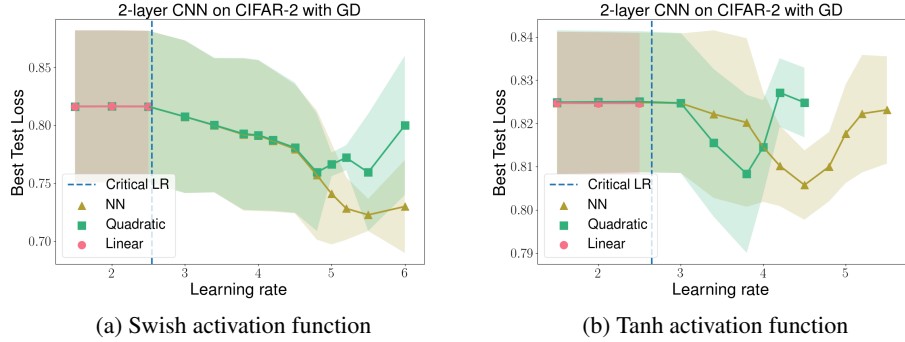

(a) Swish activation function        (b) Tanh activation function

Figure 14: **Best test loss plotted against different learning rates for** $f_{\text{quad}}$, $f$**, and** $f_{\text{lin}}$**.** We choose 2-layer CNN as the architecture and train the models on CIFAR-2 with GD.

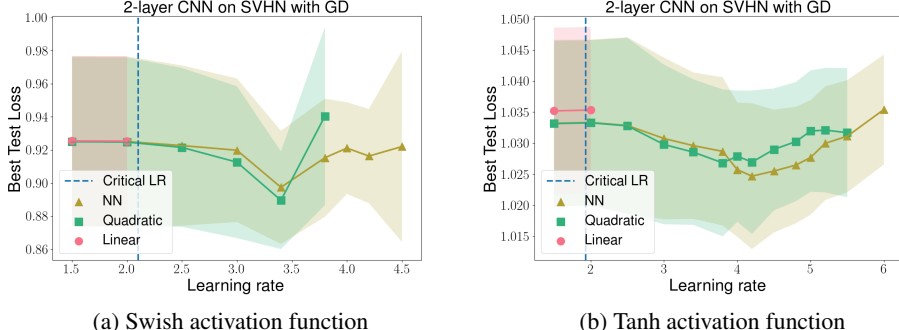

(a) Swish activation function          (b) Tanh activation function

Figure 15: **Best test loss plotted against different learning rates for** $f_{\mathrm{quad}}$**,** $f$**, and** $f_{\mathrm{lin}}$**.** We choose 2-layer CNN as the architecture and train the models on SVHN with GD.

