# OpenReview forum: "Quadratic models for understanding neural network dynamics"
_ICLR.cc/2023/Conference — Submitted to ICLR 2023_

### Official Review · Reviewer_Ctfq · 2022-10-21

**Confidence:** 5
**Correctness:** 4
**Technical Novelty And Significance:** 4
**Empirical Novelty And Significance:** 2
**Recommendation:** 6

**Clarity, Quality, Novelty And Reproducibility:**

Clarity: The paper is well-written.

Quality: The theoretical results are novel and interesting.

Originality: The analysis proposed in this article is, as far as I know, the first time the catapult phase has been established beyond linear networks.

**Strength And Weaknesses:**

Strengths:

- It is rare to be able to more or less fully analyze any kind of non-linear learning dynamics. That the authors manage to do this, even if it is for somewhat stylized datasets, in a regime of large learning rate is really nice.
- The fact that the top eigenvectors for the NTK and the Hessian provably align in the catapult phase is an interesting and important contribution. Empirically, this is often known to happen but as far as I am aware this is the first time this has been proved to occur in a non-linear model.

Weaknesses:

- The restriction to very simple datasets, either with one example, or inputs in 1D is quite severe. While the authors write “The assumption of d = 1 is common in the literature and simplifies the analysis for neural networks,” this assumption drastically simplifies (at least as far as I know) the structure of the NTK, as evidenced by Proposition 1. Perhaps the authors could comment on what is expected to occur in higher dimensions.
- To a lesser extent, the restriction to one layer networks is undesirable. Perhaps the authors could comment on what is expected to occur in deeper networks.
- There is a slightly strange apparent inconsistency between the theory developed in this article and the original work on the catapult phase. Specifically, figures 3 and 4 in that article seem to show that for shallow ReLU networks the catapult phase is delineated approximately by taking  the learning rate to be between $2 / \lambda_{max}$ and $12 / \lambda_{max}$. However, the authors here seem to prove that the largest convergent learning rate is $4 / \lambda_{max}$. Of course this is not a direct contradiction: quadratic models are not necessarily high enough fidelity to capture the constants controlling the catapult phase. Nonetheless, it would be nice if the authors could comment on this point.

**Summary Of The Paper:**

This article analyzes the dynamics of GD for quadratic approximations to sufficiently wide one layer ReLU network in which the network function is replaced by its second order Taylor expansion with respect to the parameters. The main result of this article is that for rather simple data (either one datapoint, or multiple datapoints in 1D), unlike linear approximations, quadratic approximations exhibit the so-called catapult phase. Specifically, for moderately large learning rates the training loss is unimodal, first rapidly growing and only then decaying, and the top eigenvalue of the NTK becomes decreases significantly by the end of training.

UPDATE AFTER REBUTTAL AND READING OTHER REVIEWS
I share the concerns of the other reviewers regarding the rigor of the proof of the main results. Originally I was kind of OK with it because I saw that I could fill in the details myself. But ultimately I when I read the other reviews and looked again, I became of the opinion that I quite like the results but that a paper of this kind should really have a higher standard for providing precise mathematical details in the proofs. So I am revising my assessment of this paper down from an 8 to a 6.

**Summary Of The Review:**

This article gives a novel analysis of GD with moderate learning rates in a second order approximation to wide one layer ReLU networks. The main result of this article is that for simple data (either one datapoint, or multiple datapoints in 1D), unlike linear approximations, quadratic approximations exhibit the so-called catapult phase. As mentioned in the strengths section, it is rare to give a more or less complete characterization of GD dynamics in non-linear models at relatively large learning rate. I think the basic observations and techniques in this article will be useful for further developing theory and also that the results themselves are interesting.

---

> ### Author Response · Authors · 2022-11-14
> **Response to Reviewer Ctfq**
>
> We thank the reviewer for the positive feedback.
>
> *Q: Extension of the theoretical results to multi-dimensional training examples.*
> >This is an important but seemingly quite involved question. As shown in our paper, for uni-dimensional input, the tangent kernel is at most rank 2 (Proposition 1) hence the analysis can be focused on those two corresponding eigen directions. Furthermore, due to the alignment of the Hessian and NTK for two-layer ReLU network, the second order term affects training dynamics in two eigendirections are orthogonal, resulting in clean and accurate dynamics equations.
>
> >For multi-dimensional training examples,  the rank of the neural tangent kernel can be more than $2$, hence there are more eigendirections that need to be taken into consideration. For the second order term, the alignment between the Hessian matrix $H_f$ and the feature matrix $\nabla_\mathbf{w}f$ is more difficult to check.
>
> >However, with an appropriately chosen learning rate ($2/\lambda_1<\eta<2/\lambda_2)$, we expect it may be possible to analyze the catapult phase. Since in this case, the loss increases mainly along the  eigen direction corresponding to the largest eigenvalue of NTK, i.e., $\lambda_1$. For NTK,  in general, only its largest eigenvalue has significant decrease.  As the second-order term causes the catapult phase, we believe there is a certain alignment between the top eigendirection of feature map $\nabla_\mathbf{w}f$ and the Hessian matrix. With such alignment, the effect of the second order term concentrates on the top eigendirection to cause the loss to increase and then decrease in this direction.  Therefore, certain approximated dynamics equations can be built where the change of NTK only focuses on top eigen direction, then a similar analysis as ours can be used.
>
> *Q: Extension of the theoretical results to deeper nets.*
> >We expect similar results still hold for multi-layer neural networks, although the analysis needs more advanced techniques.
> The main difference between shallow and deep nets with respect to the analysis of the catapult phase is the sparsity of the Hessian matrix of the network. In our current analysis for a two-layer network, the Hessian matrix of the network is sparse, which allows us to figure out the alignment between the Hessian and NTK. For the deep networks, the  Hessian matrix of the network becomes dense, which makes finding the alignment more difficult. However, as recently found by [1], under an appropriate rotation of coordinates in the parameter space, the Hessian matrix of the network function becomes sparse. We believe combining our analysis and this technique can help analyze the alignment and hence the catapult phase.
>
> >[1] Liu, Zhu, and Belkin. ``Transition to Linearity of Wide Neural Networks is an Emerging Property of Assembling Weak Models." ICLR 2022.
>
> *Q: Different largest convergent learning rate between our work and the original work on the catapult phase ($4/\lambda_1$ vs. $12/\lambda_1$).*
>
> >We found that the largest convergent learning rate seems to depend on the dataset and the network architecture for both neural networks and NQMs. For example, in our empirical results, Figure 4, the large convergent learning rate (close to the maximum learning rate shown in figure) seems not to be $12/\lambda_1$ and varies among different datasets and architectures. As we can see from the original catapult phase paper, the $12/\lambda_1$-rule is not a theoretical result, but an observation from a few experiments.
>
> *Q:  Whether quadratic models are high enough fidelity to capture the constants controlling the catapult phase.*
>
> >We observed in experiments that the largest convergent learning rates for networks and NQMs are different. Therefore, we don't claim that NQMs have exactly the same largest convergent learning rate as neural networks.  However,  in our theoretical analysis for multiple training examples, the constant $4$ arises from the fact that NTK has exactly two eigen directions. Therefore, it is possible that when $d>1$ and more eigen directions are involved in training, the constant will change and be closer to that of networks.

---

### Official Review · Reviewer_JdaH · 2022-10-21

**Confidence:** 2
**Correctness:** 4
**Technical Novelty And Significance:** 4
**Empirical Novelty And Significance:** 4
**Recommendation:** 8

**Clarity, Quality, Novelty And Reproducibility:**

Clarity: The paper is well written and clearly structured

Quality: The theoretical analysis is of high quality

Novelty: The presented analysis seems to be a novel, original and deeply insightful extension of the existing literature on analyzing training dynamics via the neural tangent kernel idea

Reproducability: The focus of the paper is the theory which is elaborated in rich detail in the appendix

**Strength And Weaknesses:**

Strengths:
- The paper analyzes an interesting practical phenomenon using powerful theoretical tools that will enable better understanding of training dynamics for neural networks
- Having the main theoretical analysis demonstrated for the single training example case first helps make it more accessible
- The empirical validation in Figure 3 backs up the theory

Weaknesses:
- NQMs appears in the figure caption for Figure 1 (b) which is referenced before the abbreviation (for Neural Quadratic Model) is first introduced in the text
- Does the theoretical analysis in principle extend for squared to cross-entropy loss? (Makes the dynamics less elegant but would still be interesting to know how this could work)

**Summary Of The Paper:**

The authors present the theoretical analysis of a neural quadratic model as an approximation of a two-layer neural network.
The neural quadratic model exhibits an interesting non-linear aspect of training dynamics that can be seen with two-layer neural networks in practice: The catapult phase.
The catapult phase occurs when the learning rate is high enough to be beyond locally critical but low enough so as to not globally diverge and leads to a jump into a potentially better optimum with lower
neural tangent Kernel norm.
The authors' theoretical analysis makes use of insightful estimation possible due to considering the wide network limit but the theoretically predicted phenomena are relevant in practice and the behavior in training loss and tangent kernel largest eigenvalue are also demonstrated using experiments.

**Summary Of The Review:**

I recommend to accept the paper as I perceive it to analyze an interesting practical phenomenon of neural networks training using and extending the toolbox of modern theoretical analysis tools for neural
network training dynamics.

---

> ### Author Response · Authors · 2022-11-14
> **Response to Reviewer JdaH**
>
> We thank the reviewer for the positive feedback.
>
> *Q: Unreferenced NQMs in the figure caption for Figure 1 (b).*
> >Thanks for pointing this out. We have fixed it in the revision.
>
> *Q: Extension of the theoretical analysis to cross-entropy loss.*
> >This is a good question.  To the best of our knowledge,  the catapult phase phenomenon for wide neural networks has only been consistently observed with square loss. As noted in the original catapult phase paper [2], the Hessian of the loss is not constant for cross-entropy loss hence to control experiments is more challenging.
> While [1] observed the catapult phase phenomenon on deep *linear* networks with cross-entropy loss,  it is unclear whether networks with non-linear activation functions can consistently exhibit this phenomenon as well. While this analysis goes beyond the scope of the current paper, we believe NQMs can be a useful tool to analyze the training dynamics with other losses.
>
>
> [1] Huang, W., Du, W., Da Xu, R. Y., \& Liu, C. (2020). Implicit bias of deep linear networks in the large learning rate phase. arXiv preprint arXiv:2011.12547.
>
> [2] Lewkowycz, Aitor, et al. "The large learning rate phase of deep learning: the catapult mechanism." arXiv preprint arXiv:2003.02218 (2020).

---

### Official Review · Reviewer_LYg1 · 2022-10-24

**Confidence:** 2
**Correctness:** 3
**Technical Novelty And Significance:** 3
**Empirical Novelty And Significance:** 3
**Recommendation:** 6

**Clarity, Quality, Novelty And Reproducibility:**

This paper is well-written. I did not fully check all the details of the proof though they are assumed to be right. ]
Moreover, this paper is novel in firstly analyzing the non-linear wide neural networks  the catapult regime through the perspective of the quadratic approximation

**Strength And Weaknesses:**

Strenghths:

1. It is a solid theoretical work on the learning dynamics of deep neural networks. The adopted quadratic model successfully captures the interesting "catapult phase" from Lewkowycz et al. (2020), which provides insights and thus help understanding of the sophisticated dynamics of NN.

2. Numerical results support their theoretical analysis.

Weaknesses:

1. The output dimension d is limited to dimension 1 and it is unclear what happens in the case d>1, and whether similar results still hold, which I think is the biggest weakness, since in high-dimensional case, some counter-intuitive phenomena might happen.



**Summary Of The Paper:**

This paper studies the learning dynamics of deep neural networks using the quadratic models.  Specifically, the "catapult phase" is identified under this model when the learning rate is high, while it is not for linear models. Numerical simulations are provided to support the theoretical analysis.





**Summary Of The Review:**

A solid theoretical work on the analysis of the learning dynamic of NN by using the quadratic models, as opposed to previously used linear models. The most interesting result is that, it  successfully captures the interesting "catapult phase" from Lewkowycz et al. (2020), while linear networks fail. This demonstrates the usefulness of the quadratic models in understanding NNs. The only concern is that the current analysis is restricted to the output dimension d=1 and it is not clear whether the results or methods still apply to high-dimensional outputs, or even deeper hidden layers.

---

> ### Author Response · Authors · 2022-11-14
> **Response to Reviewer LYg1**
>
> We thank the reviewer for the positive feedback.
>
> *Q: Whether similar results hold for the case when the output dimension is greater than $1$.*
> >Though our theoretical results only consider scalar output,  in our empirical results (Figure 4), we showed NQMs with output dimension $C>1$ still have catapult phase with dataset MNIST ($C=10$), SVHN ($C=10$), FSDD ($C=6$) and AG NEWS ($C=4$). Therefore, empirically, similar results still hold when the output dimension is larger than one.
>
> *Q2: `` Some counter-intuitive phenomena might happen in high-dimensional case''.*
> >To the best of our knowledge, we are not aware of such phenomena in this setting. Do you have a specific phenomenon in mind?

---

### Official Review · Reviewer_uqyZ · 2022-10-26

**Confidence:** 4
**Correctness:** 2
**Technical Novelty And Significance:** 4
**Empirical Novelty And Significance:** 4
**Recommendation:** 5

**Clarity, Quality, Novelty And Reproducibility:**

Clarity: The paper is very clear and conveys very well the intuitions behind the results. My only concern regarding the clarity is that one critical assumption is relatively "buried" in the paper (It is not mentioned as an "official" assumption and only appears in section 3.2). More precisely, for their main result (Thm 2), the authors assume that the input data is unidimensional ($x_i \in \mathbb{R}$. It is an assumption that is quite restrictive, and that should be mentioned in the abstract, introduction, and related work. In particular, the author claims that their formulation is more general than Lewkowycz et al. [2020] but it seems that Lewkowycz et al. [2020] does not have to restrict their analysis to unidimensional inputs. If it is the case, I think the author should mention it in the introduction/related work.
Quality: Except for my concern regarding the proofs of the main results of Theorem 1 and 2, I think the quality of this paper is pretty good.

Novelty: To the extent of my knowledge, the results are novel.

Reproducibility: I think that the proofs of the main theorems (Thm 1 and 2) lack details and formalism. So I am not sure an independent researcher could actually reprove these results formally on the basis of the proof sketches presented in this paper.

Minor comment:
- End of page 5 you write that $\lambda(t) = O(1)$ but actually, what you need is $\lambda(t) = \Theta(1)$ (which is what you noted elsewhere in the paper)


**Strength And Weaknesses:**

## Strengths:
- This paper tries to tackle a difficult problem which is to understand the non-linear dynamics of learning neural networks.
- The model they are proposing is simple enough to be analyzed but complex enough to model non-linear learning dynamics that seem to correspond (to a certain degree) to what happens with two-layers ReLU networks. In particular, it exhibits a 'catapult phase' for the step-size.
- The intuition of the main elements of the proofs is well communicated.
- The experiments back up the theory pretty well.

## Weaknesses:
- My first main complaint would be regarding the critical assumption regarding the fact that the input has to be of dimension 1. This is very restrictive and relatively buried in the paper (see my comments in the section about clarity)
- My second main complaint would be with the proof of the main theorem. (In particular, Theorem 1). What I  thought was a proof sketch was actually supposed to be the complete proof (please let me know if I missed the detailed proof). The statement in the "proof of Theorem 1" (page 6 and 7) are not formal enough to be verified. I give a very good idea of the structure of the actual proof, but it is not enough. We cannot formally have $g(t) = o(\sqrt{m})$ for $t<T_{incr}$ and then $g(T_{incr }) = \Theta(m)$ (even though I understand the intuition).
- The argument that $\sum_{t=0}^{T_{incr}} \Theta(g(t)^2/m) = o(1)$ is not 100% clear to me since $g(t)$ will be close to $\Theta(m)$ when $t$ is close to $T_{incr}$, moreover even if we had $g(t)^2= o(\sqrt{m})$ we would have $\sum_{t=0}^{T_{incr}} \Theta(g(t)^2/m) = o(T_{incr}) = o(\log(m))$.


**Summary Of The Paper:**

This paper proposes an approximation of Shallow ReLU networks in the large width regime to study their optimization dynamics. More particularly, they show that for uni-dimensional inputs, this approximation exhibits the 'catapult phase phenomenon', which corresponds to a regime of learning rate larger than the largest step size advocated by the theory on linear dynamics.

**Summary Of The Review:**

This paper provides some exciting contributions. However, I have some concerns regarding the formalism of the proof. The results feel correct to me, but, at this stage, the paper only contains sketches of proof. Moreover, I am slightly uncomfortable with how a critical assumption is presented. This critical assumption could be missed by quickly reading the paper. It should be mentioned earlier in the text, and it should be more visible.

For all these reasons, I am currently evaluating this paper under the acceptance threshold. However, if the authors address my concern, I am eager to significantly increase my score.

---

> ### Author Response · Authors · 2022-11-14
> **Response to Reviewer uqyZ**
>
> We thank the reviewer for the valuable and insightful feedback. We revised our paper accordingly, and we address your concerns below.
>
> *Q: ``The uni-dimension assumption on the input data is very restrictive and relatively buried in the paper''.*
> > First, we note that the original catapult phase paper [1] also assumed the input dimension $d=1$ for the two-layer linear network analysis. Furthermore, it additionally requires a single training example, while we don't have this requirement. The more general multi-dimensional analysis in [1] is an informal argument.
>
> >Note that we have an analysis in the multi-dimensional case with a single training sample. We believe our multiple training example results extend to the multi-dimensional case, although the analysis becomes much more involved. We consider our analysis with the uni-dimensional assumption as a starting point to understand the non-linear optimization dynamics, and we would like to put the multi-dimensional multi-sample case as future work.
>
> >As to the assumption being buried, thanks for pointing this out. In the revision, we have made this assumption explicit (see Assumption 1) and made it clear that our results for multiple training examples depend on this assumption in the introduction (see Main contributions).
>
> >[1] Lewkowycz, Aitor, et al.``The large learning rate phase of deep learning: the catapult mechanism." arXiv preprint arXiv:2003.02218 (2020).
>
> *Q:`` The authors claims that their formulation is more general ..  but Lewkowycz et al. [2020] does not restrict their analysis to unidimensional inputs.''*
> > If we understand the comments correctly, this question contains two separate assessments: 1), whether our formulation of the NQM contains the two-layer linear network as a special example; 2), whether our analysis generalizes the one in Lewkowycz et al. [2020].
>
> >1) For the first point about formualtion, we made this claim for our general quadratic model (Eq.(3)) that the  two-layer linear network analyzed in Lewkowycz et al. [2020] is a special case of the general quadratic model. We only discussed the case with uni-dimensional input, however even for multi-dimensional input, the general quadratic model still includes the two-layer linear network. Given an input $\mathbf{x}\in\mathbb{R}^d$, we let the off-diagonal block of $\Sigma$ be $I \otimes \mathbf{x}$ (for uni-dimensional input $x$, the off-diagonal block will be $I x$ as our originally showed), and similarly vectorize all the weight parameters,  then the general quadratic model is equivalent to the two-layer linear network with multi-dimensional input data.  We have updated the discussion in Appendix I to justify our claim. Please see the revision.
>
> >2) For the second point about the analysis, the analysis in Lewkowycz et al. [2020] for two-layer linear networks on multiple training examples with input dimension $d>1$ is an informal argument. However, we  show accurate training dynamics equations for the quadratic models, though we assumed the input dimension to be 1.
>
> *Q:  Proofs of the main theorems (Thm 1 and 2) lack details and formalism. Cannot formally have $g(t)= o(m)$ when $t<T_{\mathrm{incr}}$ and $g(T_{\mathrm{incr}}) = \Theta(m)$.*
>
> >We have revised the statements of the main theorems (Thm 1 and 2) and the corresponding proofs and now they are formal.
>
> >In revision, we separate the increasing phase, the peak phase and the decreasing phase. Specifically,  we now use four time stamps $0<T_1<T_2<T_3<T_4$ and the increasing phase is $[0,T_1]$, the peak phase is $[T_2,T_3]$ and the decreasing phase is $[T_4,\infty)$. These three phases still describe the key characteristics in the catapult dynamics and there is no sudden change of scale as before.
>
> *Q: ``The argument that  $\sum_{t=0}^{T_{\mathrm{incr}}} \Theta(g(t)^2/m) = o(1)$ is not 100\% clear to me...''.*
>
> >We note that $g(t)^2$ grows exponentially with $t$ from $O(1)$ to $o(m)$ in the increasing phase. Hence, $\{g(t)^2\}$ can be approximately considered as a ``geometric sequence''. As we know, the summation of such a sequence is of the same order as the largest term (which is the last term). Hence,  $\sum_{t=0}^{T_1} \Theta(g(t)^2/m) = \Theta(g(T_1)^2/m) = o(1)$. (Here $T_1$ is the new notation of $T_{\mathrm{incr}}$ in the previous version.

---

### Decision · Program_Chairs · 2023-01-20

**Decision:**

Reject

**Justification For Why Not Higher Score:**

The proofs are not verifiable.

**Justification For Why Not Lower Score:**

N/A

**Metareview: Summary, Strengths And Weaknesses:**

This paper proposes an approximation of Shallow ReLU networks in the large width regime to study their optimization dynamics. More particularly, they show that for uni-dimensional inputs, this approximation exhibits the 'catapult phase phenomenon', which corresponds to a regime of learning rate larger than the largest step size advocated by the theory on linear dynamics.

All the reviewers agreed that this paper makes an interesting contribution to ICLR. They agree it should eventually be published. Yet, despite the high original ratings, reviewer uqyZ convinced the other reviewers in an extensive discussion after the rebuttal that the level of details given in the proof was not sufficient to be able to verify it and was not appropriate for an ICLR publication. They pointed out some original inconsistencies in the proof that were corrected in the revision, but there were still more gaps to fill in the revision. As an example, here is a quote from them from the discussion:
> I spent a decent amount of time on their proof. In my opinion, it is still a sketch of proof. In particular, it is not clear what happens in between their phases (e.g. what happens between $T_1$ and $T_2$, do we still have that $\lambda(t)$  increases, are we sure we can bound the sequence btw $T_1$ and $T_2$? ) ?
>
> I am not saying the result is not correct, but with this amount of details, the only way to verify the result is to spend a large amount of time writing the proof oneself. I do not think it is the task of the readers to do that.


Note that reviewers LYg1 and JdaH, while being positive, had a confidence of 2. The other reviewer Ctfq who had a confidence of 5 decided to downgrade their rating from 8 to 6 after the discussion and they agreed with reviewer uqyZ that the proof should be updated in a revision before resubmitting. The authors are encouraged to make a resubmission after including a more detailed proof.